# Finite Rate Reaction Mechanism Adapted for Modelling Pseudo-Equilibrium Pyrolysis of Cellulose

**Tomás Mora Chandía**

Mechanical Engineering Department, La Frontera University (UFRO), Temuco 4811230, Chile; tomas.mora@ufrontera.cl

**Abstract:** This manuscript is related to a formulation for modelling cellulose pyrolysis with a pseudo-equilibrium approach. The objective is to model the kinetics of the cellulose pyrolysis with a semi-global mechanism obtained from the literature in order to obtain the yield and the rate of formation, mainly that of char. The pseudo-equilibrium approach consists of the assumption that the solid phase devolatilisation can be described kinetically—at a finite rate—thus preserving the competitive characteristic between the production of char and tar, while the gas phase can be described directly by means of chemical equilibrium. The aforementioned approach gives a set of ordinary, linear, and nonlinear differential equations that are solved numerically with a consistent numerical scheme (i.e., the Totally Implicit Euler method). Chemical equilibrium was solved using CANTERA coupled with a code written in MATLAB. The results showed that the scheme preserved the tar-gas competitive characteristic for cellulose pyrolysis. The gas phase was defined as a mixture of $CO_2$, $CO$, $H_2O$, $CH_4$, $H_2$, and $N_2$, showing a similar composition compared to models from the literature. Finally, the extension of the model to biomass in general is straightforward for including hemicellulose and lignin. The formulation is described in detail throughout the document in order to be replicated and evaluated for other biological components.

**Keywords:** cellulose; pyrolysis; chemical equilibrium; chemical kinetics

## 1. Introduction

The pyrolysis process is a very ancient human activity dedicated mainly to producing charcoal fuel through specifically slow pyrolysis [1]. However, due to the increasing shortage, high prices, volatilities, and environmental issues, pyrolysis of biomass and waste materials has been considered a source of chemical substances and mainly a carbon-neutral fuel source. This work is devoted to the pyrolysis of lignocellulosic biomass. Specifically, a mathematical formulation of cellulose pyrolysis in a pseudo-equilibrium approach is proposed. The main objective is to obtain, through chemical kinetics, the rate and yield of char and an approximate prediction of the gas phase composition. The latter is an effort to develop a straightforward tool to assess pyrolyser scaling rules or the process characteristic time with a procedure that takes advantage of the comprehensive empirical and theoretical developments realised so far regarding the reaction mechanisms for biomass in conjunction with chemical equilibrium.

Pyrolysis is one of the various processes belonging to a more general group called thermal conversion processes, which also includes combustion, iron ore reduction, gasification, and fine ore sintering among others [2]. If the raw material is biomass, then it is possible to define the term biomass thermal conversion and then biomass pyrolysis. Di Blasi [3] defined pyrolysis as the process in which the mass is heated above 400 °C and the solid structure starts to decompose into many different products: char, liquids, and gases. Authors such as Basu [1] and Diebold et al. [4] classified the pyrolysis process as slow and fast pyrolysis, depending on the heating rate and temperature range, often being from 300 to 650 °C. Other authors such as Moldoveanu [5] defined the pyrolysis temperature slightly

differently, starting from above 250 °C and increasing up to 800 °C. Factors such as the heating rate, particle size, type of biomass, reactor temperature, and residence time directly affect the product distribution [1].

The pyrolysis process can be divided into several sub processes (i.e., cracking, devolatilisation, carbonisation, dry distillation, destructive distillation, thermolysis, and gas phase cracking) [1]. The primary process consists of the production of char and non-condensable and condensable gases. The latter product can react to produce more non condensable gases with low molecular weight (carbon dioxide, carbon monoxide, methane, hydrogen, water, and other light gases) and lighter condensable gases through homogeneous and heterogeneous reactions.

Slow pyrolysis often named carbonisation, produces mainly char. The biomass is heated at low rate over an extended period of time in the absence of oxygen. The process generates pyrolysis gases, or pyrolysates, but due to the extended residence time, the gas phase can undergo further heterogeneous reactions to produce more char [1]. Diebold et al. [4] explained that the slow removal of volatiles from the material (to reduce mass) facilitates secondary reactions (heterogeneous reactions) between volatiles and char, thus favouring the formation of secondary char.

On the other hand, the definition and purpose of fast pyrolysis, given by Diebold et al. [6], is the thermal conversion of biomass to produce mainly condensable organic vapors (also named liquids or bio-oils) and minimise char and gas. Fast pyrolyser units can introduce an amount of air (under a stoichiometric point) to produce heat and sustain the process. This stoichiometric combustion produces water vapor and carbon dioxide.

As stated by Basu [1], the primary goal of fast pyrolysis is to maximise the production of liquids. The heating rate is considerably higher than in slow pyrolysis, reaching 10,000 °C/min. However, the maximum temperatures, as mentioned above, are often up to 800 °C. Due to the high heating rates, the biomass remains inside the reactor for less time than in slow pyrolysis.

The composition, size, shape, and physical structure of the biomass exert some influence on the pyrolysis product through their effect on the heating rate. Finer particles offer less resistance for condensable gases, which therefore escape relatively fast to the surroundings, avoiding the secondary cracking reactions. This results in a higher liquid yield. Larger particles, on the other hand, facilitate secondary cracking reactions due to the higher resistance and have been part of the ancient method for charcoal production of using stacks with large-sized wood pieces [1].

Residence time influences the process as mentioned above. If it is necessary to maximise the char yield, a longer residence time must be applied for gases in the reaction zone in order to favor secondary reactions that can convert liquids into more char. If more gas and volatiles is the objective, the residence time must be shorter, as in the order of milliseconds, and a fast quenching process after the devolatilisation is also needed in order to freeze reactions [1].

## 2. Mathematical Modelling of Pyrolysis

Pyrolysis is a very complex multi-stage process from the biomass structure and chemical points of view [1,3,4,7]. Mathematical modelling of pyrolysis is very important since it provides knowledge of fuel reactivity, a key factor for the formulation of reactor designs and scaling rules [3]. The pyrolysis process can be described by two coupled phenomena: kinetics and transport. While the former describes the thermal decomposition of materials, the latter describes the transport behaviour (i.e., mass, energy, and momentum). These two phenomena commonly show a strong coupling due to how, in order to sustain the thermal decomposition, energy must be supplied. This energy supply can be realised mainly through convection, advection, and conduction and in a lesser quantity by radiation [7].

## 2.1. Chemical Kinetics

The primary decomposition data of lignocellulosic biomass is often obtained from weight loss analysis, commonly with thermogravimetry (TG) experiments applying isothermal or dynamic conditions. TG analysis can give the data to construct global or semi-global reaction mechanisms [3]. Due to the complex structure of lignocellulosic biomass and reaction pathways, global or semi-global mechanisms are often used for each of the lignocellulosic components (i.e., cellulose, hemicellulose, and lignin). The devolatilisation process of each of these three components can be considered in parallel and independent. The latter corresponds to a hypothesis with a good experimental verification, and it is named the superimposed principle [8]. However, all the products released from the components into the gas phase participate in a competitive manner to produce the final products.

TG analysis can reveal the thermal behaviour of a sample of any kind of material susceptible to thermal decomposition. However, if the pure kinetic effect is pursued, Di Blasi [3] and Branca et al. [9] stated that heat and mass resistances must be avoided. For example, in pure cellulose pyrolysis, the levoglucosan (LVG), the main component released, is decomposed inside the particle if there is a restriction to flow out [10]. This restriction (i.e., the mass resistance-transport effect) produces more char. Therefore, the transport effect is superimposed, and subsequently, no pure kinetics is obtained. Other related factors mentioned by Di Blasi [3] are the endo-exothermicity and thermal inertia.

Reaction mechanisms can be classified by the level of detail. A one-step global mechanism includes only one reaction with one set of kinetic parameters and describes the evolution of three main products: char, liquids, and gases. However, as stated by Di Blasi [3] and Arseneau [10], pyrolysis has competitive reactions, and therefore, a one-stage and parallel reaction mechanism is more suitable, thus allowing the prediction of the distribution of products. More detailed mechanisms include the secondary cracking reactions. Secondary reactions are heterogeneous processes catalysed by char that can break down the primary vapors to produce lighter liquids, gases, and more char [1,3].

A commonly used reaction mechanism for the thermal decomposition of cellulose is the Broido–Shafizadeh scheme [11]. A one-stage mechanism with parallel reactions describes the char, liquid, and gas yields in a competitive way [12]. The modified Broido–Shafizadeh mechanism has four reactions in series and a parallel one, including degradation first, which produces activated cellulose, dehydration, depolymerisation, producing LVG (a liquid precursor), and secondary cracking reactions. Other examples are the two-stage mechanism with secondary reactions given by Diebold et al. [6] and the superimposed reaction mechanism given by Ranzi et al. [13].

## 2.2. Equilibrium Modelling

Thermodynamics is a powerful discipline that allows for predicting thermal process behaviors. Actual thermal systems seldom reach thermodynamic equilibrium due to their characteristic finite time scale, while in thermodynamics, the time scale can be infinitely large by definition. An advantage of thermodynamic analysis is that an approximate solution can be obtained with minimal input information. Therefore, for example, if the kinetics is not known, an estimate is always available. Another advantage is that equilibrium defines the limits of the process, and any non-equilibrium description cannot produce results beyond these limits.

In dealing with chemical systems, the thermodynamic equilibrium can be calculated using the entropy principle. For an adiabatic closed system, entropy can only increase as the system reaches equilibrium. Equivalently, Gibbs free energy reaches a minimum in the same process [14]. The later property (i.e., Gibbs free energy) is often used for equilibrium calculation in chemical systems.

There are two methods to assess chemical equilibrium: stoichiometric and non-stoichiometric methods. The former is based on the thermodynamic equilibrium constants $K_p$, and a set of chemical reactions is required. The equilibrium is obtained by solving a set of linear and non-linear algebraic equations, often numerically, using an iterative

solver [15]. As stated by Vonka et al. [16], this is a simple method if the number of species is reduced. The latter method has the advantage that reactions and equilibrium constants are not required. Gibbs free energy is applied in conjunction with elemental species and chemical potentials. A set of nonlinear equations is obtained, and a minimisation process of Gibbs free energy is applied, including mass restrictions, in order to obtain a solution. The solution procedure is more complex compared with the stoichiometric method, but it is more suitable in dealing with systems with a higher number of species [16].

In order to solve the chemical equilibrium, some solvers are available, mainly using the non-stoichiometric approach. The most used solver is CHEMKIN, a commercial package that has been developed mainly for kinetics but also contains an equilibrium module. Another commercial package used by researchers is ASPEN plus. Finally, one free distributed package is CANTERA [17]. The latter free software has been developed by the California Institute of Technology using the Python language. CANTERA has solvers for kinetics, fluid dynamics, and thermodynamics, useful for reactor modelling, and can be used while coupled trough the application programming interface (API) with codes written in MATLAB, C++, or Fortran. A complete technical user guide for the CANTERA package is currently available in [18].

In general, thermal decomposition described by the use of thermodynamic equilibrium models (TEMs) can be encountered in the literature, often for gasification. Here, an approximate equilibrium description can be applied in the pyrolysis, gasification, or reduction zones of down- or updraft gasifiers, since the product has enough time to reach a near-equilibrium state [19]. The application of a TEM allows a straightforward evaluation of parameters such as fuel humidity, the effect of the mass ratio of the gasifying agent (e.g., steam), and the air-to-fuel ratio (ER) for the reduction phase. In the literature, a large amount of articles applying a TEM (e.g., [20,21]), where the CANTERA and SYNGAS routine were applied for gasification, can be identified. Particularly in [21], Baratieri et al. applied the chemical equilibrium including solid elemental carbon (graphite), which is seldom observed in other works. As a general result of the application of a TEM, it is possible to mention the over- or sub-estimation of the syngas, especially in the case of methane $CH_4$. However, good qualitative results can be obtained. For example, in [22], aTEM was applied for gasification based on solar energy with steam, $CO_2$, and ZnO as gasification agents, and it has shown theoretical results with the same trend as the experiments and small deviation in the magnitude of the $H_2/CO$ ratio. Since in an actual application the process time is not always higher than the characteristic time of the chemical reactions, equilibrium cannot be reached, and intermediate chemical species (TARS) are observed and affect the study of a reactor. To address this issue, the TEM model can be modified by empirical or theoretical relationships to predict the intermediate species. An example of this formulation is presented in [23], where the effect of the ER, temperature, fuel type, moisture content, and gasifying agent are studied. In this case, an empirical relationship was applied on the conversion of carbon to TAR, improving the prediction capacity of the model.

Although thermodynamic description of thermal processes has disadvantages, these principles can be applied even in dynamic (non-equilibrium) descriptions. For example, in combustion, a partial chemical equilibrium is assumed when an intermediate species is consumed faster after an initially rapid production [15]. In this case, this species can be considered in chemical equilibrium and allow the introduction of the kinetic constant $K_c$. This kinetic constant is a function of the forward and backward kinetic coefficients, both of which are related to the thermodynamical equilibrium constant $K_p$ [15].

The elegant simplicity of thermodynamic models and their ability to produce results from basic information is diminished, given the inability to (1) predict intermediate species and (2) introduce the time variable, which is a critical variable in the development and design of gasification units. Consequently, kinetic models (KMs) are usually adequate to overcome these problems. The disadvantages of using KM lie in the increase in computational complexity and the need to have additional fundamental kinetic information. The

KM models that can be identified in the literature can be separated into those called process kinetic models (P-KMs) and full transport phenomena kinetic models (TF-KMs). P-KMs can be developed in codes devoted to simulating systems through a definition of several interconnected processes. A critical disadvantage of this type of simulation, due to the lack of description of the hydrodynamics (TF), is that the residence time is not a variable but a parameter that must be obtained from another source. Examples of the application of P-KM formulation can be found in [24,25]. In the former, the Aspen Plus software package was used to simulate the pyrolysis of Beech wood with the devolatilisation model developed by Ranzi et al. [13]. In the latter article, the authors used as a base the kinetic model reported in [13] in order to validate, with good agreement, the experimental data obtained from [26–28]. Another application of a P-KM-type formulation was applied in [29], where Aspen HYSYS was used to simulate the techno-economic evaluation of the production of gasoline and diesel from the gasification of solid waste. Dhrioua et al. [30] developed a simulation of gasification of *Prosopis juliflora* in a fluidised bed reactor in Aspen Plus. In this work, although transport phenomena have not been used, parameters such as pressure drops, minimum fluidisation, and superficial velocities have been estimated through Apsen Plus subroutines. In all of these verifications, only the pyrolysis process of biomass was evaluated without the effect of transport phenomena or reactor geometry, therefore being a process-type simulation, however with good agreement with the experimental data. The most complex, but at the same time the most complete type of simulation, is the TF-KM type. This type of simulation requires description of the hydrodynamics which can strongly couple the chemical kinetics. In the case of gasification in down- or updraft-type reactors, which are frequently simulated by means of a TEM or P-KM, this can be identified in the literature [31–33], where the kinetics is based on kinetic models, among others, in [13] and where the residence time is a variable.

In an effort to obtain more suitable results using the chemical equilibrium for the thermal decomposition of biomass, some authors have explored models combining the equilibrium and the dynamical approaches. This hybrid approach requires additional parameters apart from the thermodynamic properties, such as the elemental composition of biomass.

Gøbel et al. [34] developed a model of a multi-stage fixed bed gasifier. The gas phase was defined in chemical equilibrium, and the hydrodynamic regime was approximated with the plug flow reactor approach (i.e., without radial distribution of the variables) [15], with a negligible pressure drop, and without intraparticle transport. Kinetics were obtained experimentally from TG analysis. The results showed an overestimated prediction of the temperature inside the reactor and predicted a slightly but significantly lower heating value of the final products. Lee at al. [19] combined two existing software packages to achieve a simulation of pyrolysis—the HSC package—for chemical equilibrium and the Sandia PSR for kinetics and being designed for combustion. The approach consisted of obtaining an equilibrium composition form of HSC, starting from the elemental analysis of biomass material and then using this composition as an input in the PSR code. Since the PSR code is designed for combustion, a special set-up was applied. The HSC code produced the expected composition for chemical equilibrium (i.e., mainly $CO_2$ CO, $H_2O$, $CH_4$, $H_2$, and $N_2$). However, the PSR code could predict a significant number of species such as $C_2H_2$, $C_2H_4$, $C_2H_6$, or $C_3H_8$, species observed only in kinetics models. These models can be classified as process kinetic and thermodynamical models (P-KM-TEMs) because they apply chemical equilibrium to part of the chemical process but do not apply transport phenomena. Other articles that report simulations of the P-KM-TEM type specifically formulated for gasification and preferably for Aspen Plus can be found in [35–38] and, for pure pyrolysis, in [39].

The literature review shows that TEMs and P-KM-TEMs are used for simulations that do not include TF effects or reactor geometry. Although the P-KM-TEM formulation offers good results for simulating gasification and fast pyrolysis in up- or downdraft reactors or fixed-bed pyrolysis, respectively, it is not common in the literature to identify a general

formulation that combines TF with KMs and TEMs. The factor introduced by TF may be critical and may also influence the synthesis of chemical species, especially those controlled by transport in heterogeneous reaction environments [25]. Therefore, the incorporation of TF is always desirable in a simulation to verify the effects of hydrodynamics on fundamental characteristics such as temperature distribution and residence time, which in turn allow coupling the dynamic and geometric characteristics of a reactor. Consequently, this article describes a KM-TEM formulation for easy coupling to a TF model and which is called pseudo-equilibrium in this work.

## 3. Methodology: Pseudo-Equilibrium Mathematical Description

The objective of this work is to develop a reduced model for the devolatilisation of cellulose while considering the gas phase in chemical equilibrium. Currently, research must be a part of a more general model for a fixed-bed reactor needed to predict the yield and rate of formation of char for their use in the eco-fertiliser industry. In this application, liquid and gas substances are considered byproducts, and the main goal is to achieve the maximum amount of char possible. Therefore, it requires a compact model, for engineering purposes, to predict the thermal behaviour of the reactor and the char yield in order to assess different designs, determine scale rules, optimise the process, and develop control strategies.

From the mathematical modelling point of view, the main factors in this research are, in essence, the formation rate and the final yield of char. Other process parameters such as details about the composition of liquid and gas phases have minor importance at this initial stage. Therefore, in order to obtain a more accurate char evolution description, a suitable kinetics description must be applied. Nonetheless, the whole gas phase (i.e., gases and liquids) are also important in relation to the mass and energy balances due to there being homogeneous chemical reactions that can consume or release energy, affecting the reactor temperature. As a consequence, the whole gas phase is an important energy carrier inside the reactor and must also be accounted for. However, it is assumed that the gas phase species distribution does not considerably affect the char formation and represents a reasonable starting point in order to obtain a model for engineering purposes.

The first step in developing the reactor model is the definition of the chemical framework which is devoted, in this work, only to cellulose material. However, since it is possible to apply the superimposed approach [8], the inclusion in the model of hemicellulose and lignin components can be treated as an extension.

### 3.1. Process Description

A whole reactor model can include a field description based on the conservation of mass, energy, and momentum. However, only the mass conservation principle is required here. An idealised reactor is described in Figure 1. The process is set at a constant pressure, while the temperature can be defined as constant or a variable in time (i.e., applying a heating rate as a boundary condition). Additionally, geometry and transport characteristics have not been considered, and therefore, the gradients of temperature, concentration, and pressure do not influence the process.

As can be seen in Figure 1, the heat added to the reactor warms up the solid substrate and the gas phase and induces the exit of mass. As the devolatilisation is taking place, the density of the solid decreases, and gradually, the biomass is converted to char. Since the pressure is considered constant, an amount of the chemical species released from the solid phase must be removed from the reactor.

Under this simple reactor description, a reaction mechanism (kinetic model) is required in order to describe the yield and formation rate of char and gases. For this purpose, a two-stage parallel and semi-global mechanism for cellulose given by Ranzi et al. [13] was chosen.

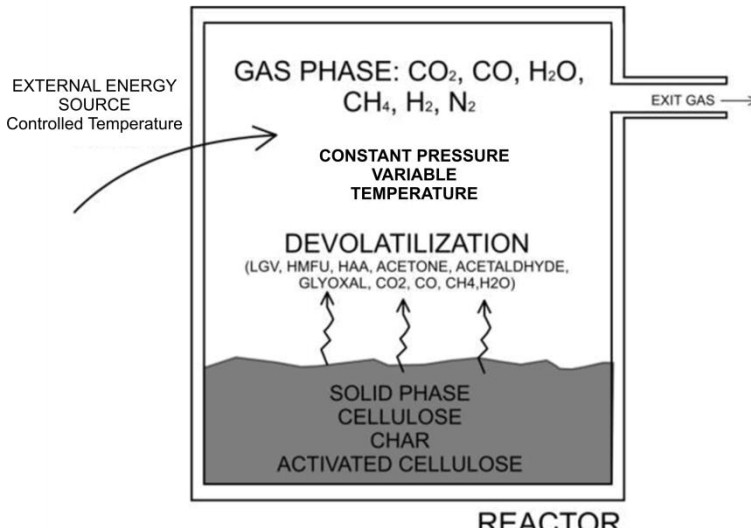

**Figure 1.** Idealised reactor and chemical interaction.

*3.2. Reaction Mechanism for Cellulose*

The reaction mechanism for cellulose developed by Ranzi et al. [13] is shown in Figure 2. This mechanism belongs to a more complete scheme that includes reactions and kinetic data for hemicellulose and lignin. This scheme has been used by Miller et al. [8] with additional reactions to describe the secondary cracking in order to develop a more comprehensive pyrolysis model, including hydrodynamics.

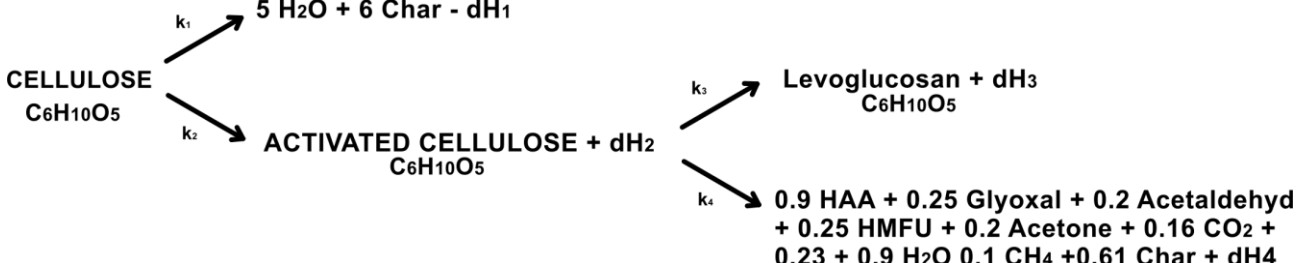

**Figure 2.** Two-stage parallel semi-global mechanism by Ranzi et al. [13].

As can be observed in Figure 2, cellulose is decomposed irreversibly and in a competitive way in activated cellulose and in parallel in both water steam and char. The later species have the same elemental composition of cellulose but with different and properties, such as formation enthalpy [13]. Both reactions have their own kinetic constants shown in Table 1, where $A$ 1/s is the pre-exponential factor, $E$ kJ/mol is the activation energy, and $T$ (-) is the temperature factor. In a consecutive manner, activated cellulose is decomposed irreversibly in LVG and in a set of gaseous species.

**Table 1.** Kinetic parameters for the reaction mechanism [13].

| Reaction | Reactant | | Products | A (1/s) | E (kJ/mol) |
|---|---|---|---|---|---|
| R1 | Cellulose | ==> | Activated Cellulose | $8 \times 10^{13}$ | 192.5 |
| R2 | Cellulose | ==> | 5 $H_2O$ + 6 Char | $8 \times 10^{7}$ | 125.5 |
| R3 | Activated Cellulose | ==> | LVG | 4T | 41.8 |
| R4 | Activated Cellulose | ==> | 0.95 HAA + 0.25 Glyoxal + 0.2 Acetaldehyde + 0.25 HMFU + 0.2 Acetone + 0.16 $CO_2$ + 0.23 CO + 0.9 $H_2O$ + 0.1 $CH_4$ + 0.61 Char | $1 \times 10^{9}$ | 133.9 |

An important chemical species released by the thermal decomposition of cellulose is LVG, and as stated by Arseneau [10], it can be released as a gas and can also participate in intraparticle reactions catalysed by char to produce different non-condensable gases.

Table 1 shows the details of the reactions and their kinetic parameters. It can be observed that the activation energy value for R1 was higher, producing activated cellulose and gases. It can also be observed that the activation energy for R2 was significantly lower. Therefore, for lower temperatures, the char yield was favoured. This is an advantage of the parallel semi-global mechanism: the competitive factor.

### 3.3. Mechanism Adaptation

In order to consider the gas phase to be in chemical equilibrium, thermodynamic data are needed for all species. The software package CANTERA, used in this work, contains a large set of thermo-chemical data for common species. However, species such as HAA, Glyoxal, Acetaldehyde, and HMFU are not present in the database. One solution is to obtain all the thermodynamic data for from Shomate or NASA polynomials. However, since the equilibrium solver of CANTERA is based on the non-stoichiometric method, the gas phase equilibrium can be solved directly with the elemental composition. Therefore, and for this purpose, the elemental composition of the gas phase must be known (with any other two thermodynamical variables) over the whole simulation. As can be observed in Figure 2, species from the solid phase were released as the devolatilisation took place, and therefore, the elemental relative composition changed constantly. In the next section, a procedure is proposed to account for this change based on the species conservation equations.

This model is a multiphase scheme, as there are two phases. Both phases are defined by a volumetric fraction because no hypotheses are made about the shape of the biomass particles. The solid phase is composed of a mixture of cellulose, activated cellulose, and char, each characterised by its density, although the total solid volume fraction, as with the gas volume fraction, is considered constant.

The species LVG, HAA, Glyoxal, Acetaldehyde, HMFU, Acetone, $H_2O$, and the other gases in R1–R4 are considered to be released in a gas state from the solid phase (as can be seen in Figure 1). However, these substances are not defined in the gas phase because it is assumed here, as a central proposition of the article, that they will be converted infinitely fast into non-condensable gases, reaching the equilibrium composition. On the other hand, char formed from R2 and R4 is considered to belong only to the solid phase. From the mathematical calculation point of view, the approach used here consists of converting the total substances released from the solid phase into an equivalent system composed solely of elemental carbon C, elemental hydrogen H, elemental nitrogen N, and elemental oxygen O in order to calculate the chemical equilibrium. The amount of gas released from the solid phase will have the same elemental composition and the same total enthalpy (an adiabatic step), although with different temperatures and pressures. This approach does not affect the final chemical equilibrium due to the unique characteristic solution of the equation system.

As a consequence of the proposition, the species LVG, HAA, Glyoxal, Acetaldehyde, HMFU, and Acetone do not belong to any phase. This process can be described as follows. The species LVG, HAA, Glyoxal, Acetaldehyde, HMFU, and Acetone are produced in the solid–gas interface and react infinitely fast to produce the species defined in the gas phase; in other words, it can be viewed as these species only existing in the interface and having infinitely short lifetimes. In addition, the elemental composition does not belong to any phase and is only a chemi-mathematical manipulation to obtain the chemical equilibrium.

### 3.4. Devolatilisation Model

Due to the zero-dimensional characteristic of the model (not accounting for field effects), the velocity field and momentum conservation equations are not required. Additionally, the energy conservation equation is possible to use to know the energy requirements

to sustain the process. However, this will be studied later in a subsequent article. Therefore, only the mass conservation equation, one for each phase, is required.

3.4.1. Mass Conservation for Solid Phase

Two volumetric phases are defined in which an integral average of the properties is applied, corresponding to a Euler–Euler approach [40]. Since, in this approach, a surface is not defined, it is not possible to apply a flux term between the phases in the conservation equations. Therefore, the mass exchange is defined through a source term. The mass conservation equation for the solid phase can be defined as

$$\frac{\partial \rho_s}{\partial t} = \dot{S}_{devo} \tag{1}$$

where $\rho_s$ is the total solid phase density and $\dot{S}_{devo}$ is the rate of devolatilisation. The term $\dot{S}_{devo}$ can be calculated from the species conservation equation [15]:

$$\frac{\rho_s Y_i}{\partial t} = \dot{S}_{devo,i} \tag{2}$$

Since we have [14]

$$\sum \frac{\rho_s Y_i}{\partial t} = \sum \frac{d\rho_i}{dt} = \frac{\partial \rho_s}{\partial t} \tag{3}$$

then we also have

$$\sum \dot{S}_{devo,i} = \dot{S}_{devo} \tag{4}$$

Considering the following irreversible reaction

$$A ==> \nu_B B + \nu_C C \tag{5}$$

then a rate equation is defined for B [41+1] as follows on a molar basis:

$$\frac{d[B]}{dt} = -\nu_B \frac{d[A]}{dt} \tag{6}$$

Ranzi et al. [13] specified that the rate of reaction to be defined by a first-order Ahrrenius-type model, and the second Raye term in Equation (6) can be calculated by the following on a molar basis as well:

$$\frac{d[A]}{dt} = -k[A] \tag{7}$$

where $k$ is the kinetic constant defined by the modified Arrhenius equation:

$$k(T) = A_0 T^n exp(-E/RT) \tag{8}$$

where $A_0$ is the pre-exponential factor, $E$ is the activation energy, and $T^n$ is the temperature factor. The superscript $n$ is the temperature factor introduced to improve the ordinary Arrhenius equation. The concentration $[A]$ in Equation (7) can be the cellulose material or activated cellulose, depending on the reaction mechanism (Table 1).

Equations (6) and (7) can be expressed on a mass basis by multiplying them by their molar masses:

$$M_B \frac{d[B]}{dt} = \frac{d\rho_B}{dt} = -\nu_B M_B \frac{d[A]}{dt} \tag{9}$$

$$M_A \frac{d[A]}{dt} = \frac{d\rho_A}{dt} = -k M_A [A] = -k\rho_A \tag{10}$$

Therefore, any products in the mechanism can expressed by combining the generic Equations (9) and (10):

$$\frac{d\rho_B}{dt} = -\nu_B \frac{M_B}{M_A}\frac{d\rho_A}{dt} \tag{11}$$

Since some species can be produced in more than one reaction, a more general expression is given by

$$\frac{d\rho_B}{dt} = -\sum_{i=1}^{M} \nu_{B,i} \frac{M_B}{M_i}\frac{d\rho_i}{dt} \tag{12}$$

The solid phase is defined by the cellulose, activated cellulose, and char. Therefore, by definition, Equation (3) can be expressed as the sum of each component:

$$\frac{\partial \rho_{Cell}}{\partial t} + \frac{\partial \rho_{ACell}}{\partial t} + \frac{\partial \rho_{Char}}{\partial t} = \dot{S}_{devo} \tag{13}$$

Equation (13) can be solved using Equation (7) when applied for cellulose and activated cellulose and with (12) for char. All the other quantities computed from Equation (12) are the total mass released in the gas phase.

Applying Equation (10) for cellulose gives

$$\frac{d\rho_{Cell}}{dt} = -(k_1 + k_2)\rho_{Cell} \tag{14}$$

and for the activated cellulose, it gives

$$\frac{d\rho_{ACell}}{dt} = k_1\rho_{Cell} - (k_3 + k_4)\rho_{Cell} \tag{15}$$

The molar mass ratio disappears on the first term on the right-hand side because the molar masses of cellulose and activated cellulose are equal. Evidently, the introduction of the activated cellulose species allows a delay in the rate of devolatilisation that strongly depends on the temperature. On the other hand, the rate formation of char is obtained from Equation (12):

$$\frac{d\rho_{Char}}{dt} = 6k_2\rho_{Cell} + 0.61k_4\rho_{ACell} \tag{16}$$

All the other rates are obtained by applying Equation (12), while the kinetics constants $k$ are obtained from Table 1 and used in Equation (8). A very useful (for computational calculation) way to obtain the yields of gaseous species is applying the form given by Turns [15]:

$$\dot{\omega}_j = \sum \nu_{ji}q_i \tag{17}$$

$$\nu_{ji} = \left(\nu''_{ji} - \nu'_{ji}\right) \tag{18}$$

$$q_i = k_{fi}\prod [X_j]^{\nu'_{ji}} - k_{bi}\prod [X_j]^{\nu''_{ji}} \tag{19}$$

where $\nu'_{ji}$ and $\nu''_{ji}$ are two *MxN* matrices (see Tables 2 and 3), *M* is the number of reactions, and *N* is the number of species in the mechanism. $\nu'_{ji}$ is the matrix that contains the stoichiometric coefficients for the reactants, and $\nu''_{ji}$ also contains the stoichiometric coefficients of products, while $[X_j]$ represents the instantaneous (in time) molar fraction. This equation can be expressed on a mass basis by multiplying them by the respective molar masses. Equations (12) and (17)–(19) produce the same set of equations as Equations (14)–(16).

**Table 2.** Reactants matrix $\nu'_{ji}$ for Equation (18).

| Reaction | Cell | ACell | Char | LVG | HAA | Glyoxal | Acetaldehyde | HMFU | Acetone | $CO_2$ | CO | $H_2O$ | $CH_4$ |
|---|---|---|---|---|---|---|---|---|---|---|---|---|---|
| **1** | 1 | 0 | 0 | 0 | 0 | 0 | 0 | 0 | 0 | 0 | 0 | 0 | 0 |
| **2** | 1 | 0 | 0 | 0 | 0 | 0 | 0 | 0 | 0 | 0 | 0 | 0 | 0 |
| **3** | 0 | 1 | 0 | 0 | 0 | 0 | 0 | 0 | 0 | 0 | 0 | 0 | 0 |
| **4** | 0 | 1 | 0 | 0 | 0 | 0 | 0 | 0 | 0 | 0 | 0 | 0 | 0 |

**Table 3.** Products matrix $\nu''_{ji}$ for Equation (18).

| Reaction | Cell | ACell | Char | LVG | HAA | Glyoxal | Acetaldehyde | HMFU | Acetone | $CO_2$ | CO | $H_2O$ | $CH_4$ |
|---|---|---|---|---|---|---|---|---|---|---|---|---|---|
| **1** | 0 | 1 | 0 | 0 | 0 | 0 | 0 | 0 | 0 | 0 | 0 | 0 | 0 |
| **2** | 0 | 0 | 6 | 0 | 0 | 0 | 0 | 0 | 0 | 0 | 0 | 5 | 0 |
| **3** | 0 | 0 | 0 | 1 | 0 | 0 | 0 | 0 | 0 | 0 | 0 | 0 | 0 |
| **4** | 0 | 0 | 0.61 | 0 | 0.95 | 0.25 | 0.2 | 0.25 | 0.2 | 0.16 | 0.23 | 0.9 | 0.1 |

The formulation detailed above produces a system of ordinary nonlinear differential Equations (12) and (17)–(19) that must be solved simultaneously.

3.4.2. Mass Conservation Equation for the Gas Phase

Whole mass conservation balance for the gas phase is a restriction in the equilibrium computations in the CANTERA software package. The mass conservation balance for the gas phase is formally presented in Equation (20):

$$\frac{\partial \rho_g}{\partial t} = \dot{S}'$$

(20)

Equation (20) does not account for the exit of mass. Since an exit surface is not defined, it is not possible to define an exit velocity, and therefore, the exit mass must be included in the source term. Consequently, we have

$$\dot{S}_{devo} \neq -\dot{S}'$$

(21)

where $\dot{S}'$ is the mass source of the gas phase. The gas phase composition is computed using the mass species conservation:

$$\frac{\partial \rho'_i}{\partial t} = \dot{S}'_i$$

(22)

Prime is added in the density of species *i* because Equation (22) is not the effective mass conservation of the whole gas phase and instead is an intermediate step only for the purpose of determining the change in elemental composition. In Equation (22) $\dot{S}'_i$ is the source term related to the solid phase. It is worth noting that $\dot{S}'_i$ is a density generation term on a solid volume fraction basis, and it must be related to the gas volume fraction *g*.

Species produced in the solid phase are released in the gas phase, but the species (LVG, HAA, Glyoxal, Acetaldehyde, HMFU, and Acetone) produced do not belong to this phase, and as proposed earlier, this amount of mass must be transformed to an elemental basis. The amount of mass released from the solid phase is known through a set of equations of the form of Equation (11). In order to transform the set of species released from the solid phase to an elemental basis, Equation (23) is defined:

$$[CHON_s]\left\{\frac{d\rho'_B}{dt}\right\} = \left\{\frac{d\rho_{El}}{dt}\right\}$$

(23)

In this Equation, $CHON_s$ is a *p x q* matrix (Table 4) that contains the conversion factor needed to obtain the amount of mass of each elemental species released from the solid phase. The latter can be recognised as an *elemental operator* that transforms a set of species to its

elemental basis. In the $CHON_s$ matrix, $p$ is the number of elemental species considered (i.e., carbon C, hydrogen H, oxygen O, and nitrogen N in this work), and $q$ is the total number of species released in the gas phase (i.e., LVG, HAA, Glyoxal, Acetaldehyde, HMFU, Acetone, $H_2O$, $CO_2$, CO, $H_2O$, and $CH_4$) (obtained from the reaction mechanism). On the other hand, $\left\{ \frac{d\rho_B}{dt} \right\}$ is the derivative of the species vector from the solid phase, and $\left\{ \frac{d\rho_{El}}{dt} \right\}$ is the resulting derivative of the elements vector. Combining Equations (12) and (23) gives

$$\left\{ \frac{d\rho_{El}}{dt} \right\} = -\frac{s}{g}[CHON_s]\left\{ \sum_{i=1}^{M} \nu_{j,i}\frac{M_j}{M_i}\frac{d\rho_i}{dt} \right\} \qquad (24)$$

**Table 4.** CHONs matrix.

|   | LVG | HAA | Glyoxal | Acetaldehyde | HMFU | Acetone | $CO_2$ | CO | $H_2O$ | $CH_4$ |
|---|---|---|---|---|---|---|---|---|---|---|
| **C** | 0.444 | 0.40 | 0.414 | 0.55 | 0.57 | 0.62 | 0.27 | 0.43 | 0.00 | 0.75 |
| **H** | 0.062 | 0.07 | 0.034 | 0.09 | 0.05 | 0.10 | 0.00 | 0.00 | 0.11 | 0.25 |
| **O** | 0.494 | 0.53 | 0.552 | 0.36 | 0.38 | 0.28 | 0.73 | 0.57 | 0.89 | 0.00 |
| **N** | 0.000 | 0.00 | 0.000 | 0.00 | 0.00 | 0.00 | 0.00 | 0.00 | 0.00 | 0.00 |

Equation (24) includes the factor $s/g$ to transform from the solid density basis to the gas density basis, where $s$ is the volumetric fraction of solid phases and $g$ is the volumetric fraction of the gas phase. It is worth noting that the variation of elements in the gas phase is expressed by the change in the cellulose and activated cellulose through $\frac{d\rho_i}{d_t}$. The final step is to compute the element mass fraction using Equation (2):

$$\frac{d\rho'_{el}}{dt} = \frac{d\left(\rho'_g Y_{el}\right)}{dt} \qquad (25)$$

In order to compute the mass fraction vector, it is necessary to calculate the term $\rho'_g$. As mentioned earlier, prime indicates that this gas density is not the equilibrium density, and it only states the change in density due to the amount of gas released from the solid phase that is added to the actual gas density. In other words, $\rho'_g$ is an "*unbalanced gas density*". This term is computed using Equations (3), (4), and (22):

$$\frac{\partial \rho'_g}{\partial t} = -\frac{s}{g}\dot{S}_{devo} \qquad (26)$$

This equation states that the variation in the solid density is equal to the unbalanced density variation of the gas phase, considering the volumetric factor conversion $s/g$. In summary, in order to obtain a solution, 14 ordinary differential equations must be solved: 3 for solid species, cellulose (Equation (14)), activated cellulose (Equation (15)), and char (Equation (16)), 1 equation for each gas released (LVG, HAA, Glyoxal, Acetaldehyde, HMFU, Acetone, $H_2O$, $CO_2$, CO, and $H_2O$ with Equation (12)), and 1 equation for the unbalanced density (Equation (26)).

*3.5. Numerical Solution*

The equation system is solved numerically with as simple a a method as possible at this starting stage, leaving a more complex solution scheme for future works. The simplest solution is provided by the Euler method, but the latter has the disadvantage that it can produce mass inconsistencies (i.e., negative densities). Applying the Euler method to Equation (15) yields the following numeric scheme:

$$\rho_{Cell\ n+1} = \rho_{Cell\ n} - \Delta t(k_1 + k_2)\rho_{Cell\ n} \qquad (27)$$

In order to obtain consistent positive solutions (i.e., the density cannot be less than zero), a restriction on the time step must be imposed:

$$\Delta t < \frac{1}{|(k_1 + k_2)|} \tag{28}$$

Since $k_1$ and $k_2$ are always positive and can be significantly large because of pre-exponential factor $A$ (see Table 1), the time step can be prohibitively small. This constraint is avoided if the totally implicit Euler method (TIE) in Equation (29) is applied. Applying the TIE to cellulose yields

$$\rho_{Cell\ n+1} = \rho_{Cell\ n} - (k_1 + k_2)\rho_{Cell\ n+1}\Delta t \tag{29}$$

Solving for $\rho_{Cell\ n+1}$ yields

$$\rho_{Cell\ n+1} = \frac{\rho_{Cell\ n}}{1 + (k_1 + k_2)\Delta t} \tag{30}$$

Since all the terms to the right of Equation (31) are positive, the scheme is consistent, always being greater than zero for any time step size. The TIE is applied to all equations except Equation (24). The later equation will be integrated later, considering the reactor volume fraction.

Applying the TIE scheme to Equations (14)–(16) gives the following set of algebraic equations:

$$Cell_{\ n+1} = \frac{Cell_n}{1 + (k_1 + k_2)\Delta t} \tag{31}$$

$$ACell_{\ n+1} = \frac{ACell_{\ n} + \Delta t Cell_{\ n+1}}{(1 + k_3 + k_4)\Delta t} \tag{32}$$

$$Char_{\ n+1} = Char_{\ n} + (6k_2 Cell_{\ n+1} + 0.61k_4 ACell_{\ n+1})\Delta t \tag{33}$$

For the species released in the gas phase, a direct integration is computed for Equation (24):

$$\int \left\{ \frac{d\rho_{El}}{dt} \right\} dt = -\frac{s}{g}[CHON_s] \int \left\{ \sum_{i=1}^{M} v_{j,i} \frac{M_j}{M_i} \frac{d\rho_i}{dt} \right\} dt \tag{34}$$

Using the fundamental theorem of integral calculus (FTIC) yields

$$\left\{ \Delta\rho_{El,n+1} \right\} = -\frac{s}{g}[CHON_s] \left\{ \sum_{i=1}^{M} v_{j,i} \frac{M_j}{M_i} \Delta\rho_i \right\} \tag{35}$$

where the integration constant is not considered because only the change in elemental concentration is required. In Equation (35), $\Delta\rho_i$ is the variation of cellulose and activated cellulose densities computed using Equations (30) and (32).

Equation (26) is also directly integrated in time. When combined with Equation (13), we have

$$\int \frac{\partial \rho_g'}{\partial t} dt = -\frac{s}{g} \int \frac{\partial \rho_s}{\partial t} dt \tag{36}$$

Using the fundamental theorem of integral calculus (FTIC) yields

$$\rho_{g,n+1}' = \rho_{g,n} - \frac{s}{g}\Delta\rho_s \tag{37}$$

where $\frac{s}{g}(\Delta\rho_s)$ is the sum of the density variation (unbalanced) of all species in the gas phase that can be calculated from Equations (31)–(33), and $\rho_{g,n}$ is the "*balanced*" density of the gas phase in time $t_n$. Therefore, the prime can be eliminated.

Equation (22) is integrated in vector form, and recognizing that

$$\rho'_i = \rho'_g Y_{g,i} \tag{38}$$

then yields

$$\left\{ \int \frac{d\rho'_g Y_{g,i}}{dt} dt \right\} = \left\{ \int \dot{S}'_i dt \right\} \tag{39}$$

Solving with the FTIC yields

$$\left\{ \left( \rho'_g Y_{g,i} \right)_{n+1} \right\} = \left\{ \left( \rho'_g Y_{g,i} \right)_n \right\} + \{\Delta S'_i\} \tag{40}$$

The last term in Equation (40) is the mass variation vector (on a volume basis) of the species considered in the gas phases (i.e., $CO_2$, $CO$, $H_2O$, $CH_4$, $H_2$, and $N_2$ for this work), and it must to be equal to the mass released from the solid phase in the interval $dt$. The first term on the right side of Equation (40) is the initial vector of species of the gas phase in $t_n$, so the prime can be eliminated. The term on the left side is the new composition of the unbalanced species densities. Equation (40) can be written to recognise that densities are scalar quantities:

$$\rho'_{g,n+1} \{Y_{g,i,n+1}\} = \rho_g \{Y_{g,i,n}\} + \{\Delta S'_i\} \tag{41}$$

The last term on the right-hand side of Equation (41) is not possible to compute due to there being no reactions nor kinetics defined to transform the species released from the solid phase to the gas phase. However, Equation (35) gives the same change in density but on an elemental basis obtained from the solid phase. Therefore, Equation (41) must be transformed to an elemental basis in order to be coupled with Equation (35) in the same dimension.

The same procedure used for Equation (23) is applied for Equation (41). Nonetheless, the *CHON* matrix is different due to the species considered in the gas phase being different from species released from the solid phase. Therefore, a new *CHON* matrix, named $CHON_g$ here, is defined (see Table 5). Applying the elemental operator to Equation (41) gives

$$\rho'_{g,n+1} \left\{ Y_{El,g,i,n+1} \right\} = CHON_g \left( \rho_g \{Y_{g,i,n}\} + \{\Delta S'_i\} \right) \tag{42}$$

**Table 5.** CHON$_g$ matrix.

|   | CO$_2$ | CO | H$_2$O | CH$_4$ | H$_2$ | N$_2$ |
|---|---|---|---|---|---|---|
| **C** | 0.273 | 0.429 | 0.000 | 0.750 | 0.000 | 0.000 |
| **H** | 0.000 | 0.000 | 0.111 | 0.250 | 1.000 | 0.000 |
| **O** | 0.727 | 0.571 | 0.889 | 0.000 | 0.000 | 0.000 |
| **N** | 0.000 | 0.000 | 0.000 | 0.000 | 0.000 | 1.000 |

Now, the vector term on the left-hand side is the unbalanced elemental vector of the gas phase. Distributing the elemental operator *CHONg* and recognizing that

$$CHON_g \{\Delta S'_i\} = \{\rho_{El,n+1}\} \tag{43}$$

gives

$$\rho'_{g,n+1} \left\{ Y_{El,g,i,n+1} \right\} = CHON_g \rho_g \{Y_{g,i,n}\} + \{\rho_{El,n+1}\} \tag{44}$$

Solving for the new elemental vector gives the new elemental mass fraction of the gas phase due to the gas released through devolatilisation process, which is

$$\left\{ Y_{El,g,i,n+1} \right\} = \frac{CHON_g \rho_g \{Y_{g,i,n}\} + \{\rho_{El,n+1}\}}{\rho'_{g,n+1}} \tag{45}$$

In other words, Equation (45) expresses that the mass fraction vector of the gas phase in time $t_n$ is converted to an elemental basis and is added to the species released from the solid phase, which is also on an elemental basis. The product with the inverse of the unbalanced density $\rho'_{g,n+1}$ gives the new mass fraction for the gas phase due to the devolatilisation process.

The vector on the left-hand side of Equation (45), in conjunction with the pressure and the current temperature, are the inputs for CANTERA to compute the equilibrium composition.

It is worth noting that the unbalanced term will disappear because the equilibrium calculation will generate a new gas density according to the new elemental vector, pressure, and current temperature. This means that there is a relaxation process in which, after the gas is released from the solid phase, the increased pressure causes the exit of mass until a constant pressure is reached again. This process is infinitely fast, and therefore, at the process time scale (macro scale), the pressure is captured as being constant. The new composition is expressed in terms of the gas phase species (i.e., $CO_2$, $CO$, $H_2O$, $CH_4$, $H_2$ and $N_2$).

*3.6. Algorithm*

In order to determine the new equilibrium composition, all the Equations of the system must be solved simultaneously. However, due to the structure of the system, it is possible to solve it in a sequential manner. Figure 3 shows a diagram for the solving procedure.

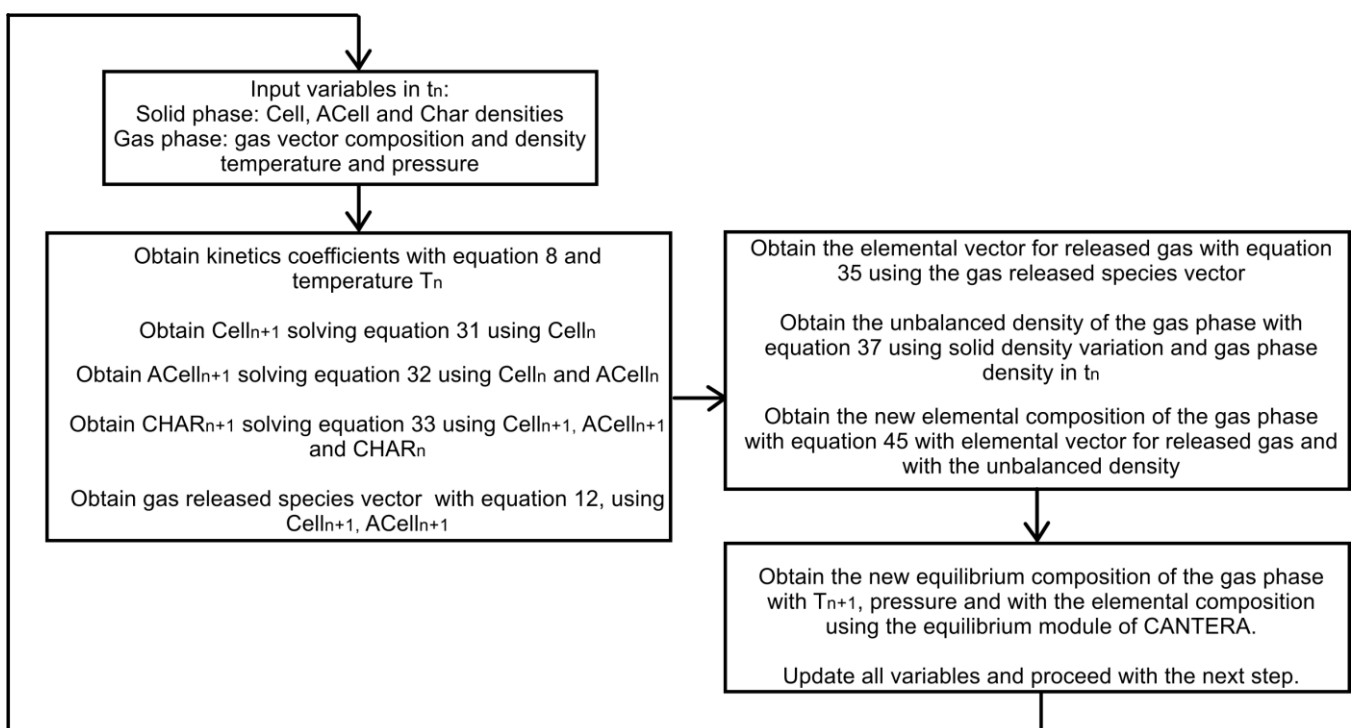

**Figure 3.** Algorithm to solve the pseudo-equilibrium problem.

## 4. Results and Discussion

The results of the model proposed in above sections are presented here. The simulation was performed by applying a linear increment of the temperature profile (constant heating rate of 1.8, 8.0, and 18.0 K/min), which is a critical controlled variable in thermal conversion processes. It is worth noting that in an actual reactor environment, the temperature profile depends on the transport properties. For example, for a fixed bed batch reactor where the biomass is in a steel box opened from the top, heat can be supplied from electrical heaters directly from the top by radiation and through the steel walls through convection and

conduction. As a consequence, the biomass temperature can be controlled linearly at the top and near the walls. However, the probable inner points in the bed do not develop linear heating due to the transport resistances. This model does not account for transport terms, so all biomass was heated up under the same regime.

Figure 4 shows the evolution of the solid phase with a heating rate of 18 K/min and starting from 300 K. The gas density in this figure is related to the gas released relative to the solid density basis. The devolatilisation process started at around 453 K (i.e., around 550 s). The species produced were mainly $H_2O$ and char through R2 directly from virgin cellulose.

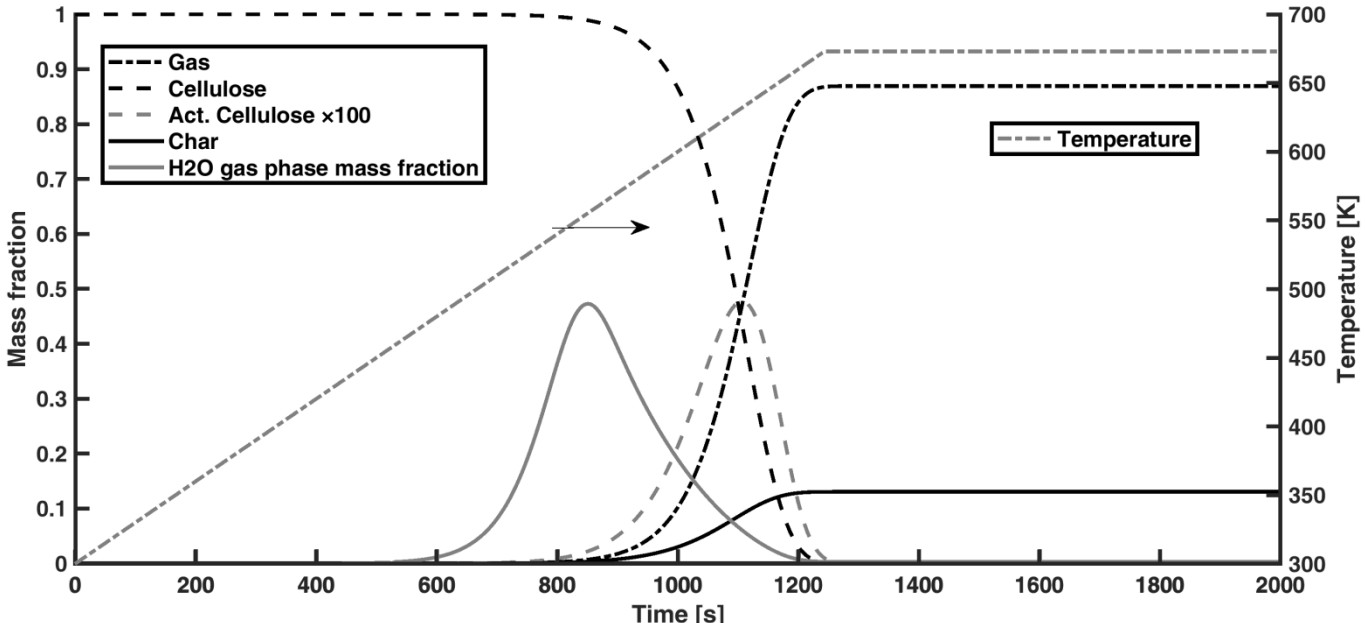

**Figure 4.** Solid phase parameters. Heating rate = 18 (K/min), $T_0$ = 300 (K), and $T_{max}$ = 673 (K).

The required properties for cellulose, activated cellulose, char, and gaseous species are tabulated in Tables 6 and 7. For all the simulations, molecular nitrogen was considered the only species in the gas phase at t = 0 due to it being a common condition in actual laboratory processes in order to avoid combustion. The initial conditions and other required parameters are also presented in Table 8.

**Table 6.** Solid physical properties.

|  | Density (kg/m³) | Molar Mass (kg/kmol) |
|---|---|---|
| Cellulose [41] | 400 | 162 |
| Activated Cellulose | 400 | 162 |
| Char (C Graphite) | 2333 | 12 |

**Table 7.** Molecular formula and molar mass [13].

|  | LVG | HAA | Glyoxal | Acetaldehyde | HMFU | Acetone | CO₂ | CO | H₂O | CH₄ |
|---|---|---|---|---|---|---|---|---|---|---|
| Molecular Formula | $C_6H_{10}O_5$ | $C_2H_4O_2$ | $C_2H_2O_2$ | $C_3HCO$ | $C_6H_6O_3$ | $C_3H_6O$ | - | - | - | - |
| Molar Mass (kg/kmol) | 162 | 60 | 58 | 44 | 126 | 58 | 44 | 28 | 18 | 16 |

**Table 8.** Initial conditions and process parameters.

| Initial Density | | | |
|---|---|---|---|
| **Solid Phase** | | | |
| Cellulose Density | | 400 | kg/m$^3$ |
| Activated Cellulose | | 0 | kg/m$^3$ |
| Char | | 0 | kg/m$^3$ |
| **Gas Phase** | | | |
| Gas species | Only $N_2$ | | |
| Pressure | 101,325 | Pa | (constant) |
| Temperature $T_0$ | 300 | K | |
| Max temperature Tmax | 673 | K | |
| Density | According to initial temperature $T_0$ and pressure | | |
| **Volume Fraction** | | | |
| Solid (s) | 0.3 | | |
| Gas (g) | 0.7 | | |
| **Heating Rate** | | | |
| Rate 1 | 1.8 | K/min | (constant) |
| Rate 2 | 9.0 | K/min | (constant) |
| Rate 3 | 18.0 | K/min | (constant) |

Near t = 700 s, instantaneous gas released from the solid phase was practically only $H_2O$, and as can be observed in Figure 5, the gas phase was therefore mainly composed only of $H_2O$ and $N_2$. LVG and other gases produced by R4 were practically not released, and, due to the activated cellulose, would start to produce only after 700 s (Figure 4). Therefore, char was produced before other gases. The quantity of gas and char produced in this first stage was relatively low due to the mass fraction relative to the solid phase and virgin cellulose largely being the main component (Figure 4). The $H_2O$ observed in the gas phase and relative to gas phase density was mainly produced by dehydration of virgin cellulose through R2. When the activated cellulose was produced, an increase in the $CO_2$ mass fraction was observed in the gas phase (Figure 5), meaning that some LGV, HAA, Glyoxal, Acetaldehyde, HMFU, $CO_2$, CO, $H_2O$, and $CH_4$ species started to be produced by the reactions R3 and R4. The latter reaction produced some water too, so part of the water present in the gas phase was due the contributions of R3 and R4. It is worth noting that the gas phase composition was not a direct consequence of the mass released from the solid phase. Instead, the distribution of gaseous species was a direct consequence of the chemical equilibrium, which was unambiguously fixed given the distribution of elemental species and the pressure and instantaneous temperature (101,325 Pa and 570 K, respectively). At these thermodynamical conditions, $H_2O$ and $CO_2$ is favoured by chemical equilibrium.

When activated cellulose was produced, the gas density started to increase rapidly (Figure 4). This was the main weight loss along the process, but due to the inertia effect caused by the kinetics (introduced by the addition of the activated cellulose [11]), the gas density started to rise after the cellulose consumption. This is clear evidence of the inclusion of a delay element in the model.

Starting from 550 K (i.e., near 900 s), competitive char–gas reactions favors the gas yield (Figure 4) according to Di Blasi's [3] observations. The pyrolysis process finished, and slightly after that, the system temperature was constant at 673 K, when the cellulose was completely depleted. This behaviour is in agreement with that described in the literature, stating that the cellulose decomposes over a range from 598 to 648 K [3], and as can be observed here, the main devolatilisation process took place near this range.

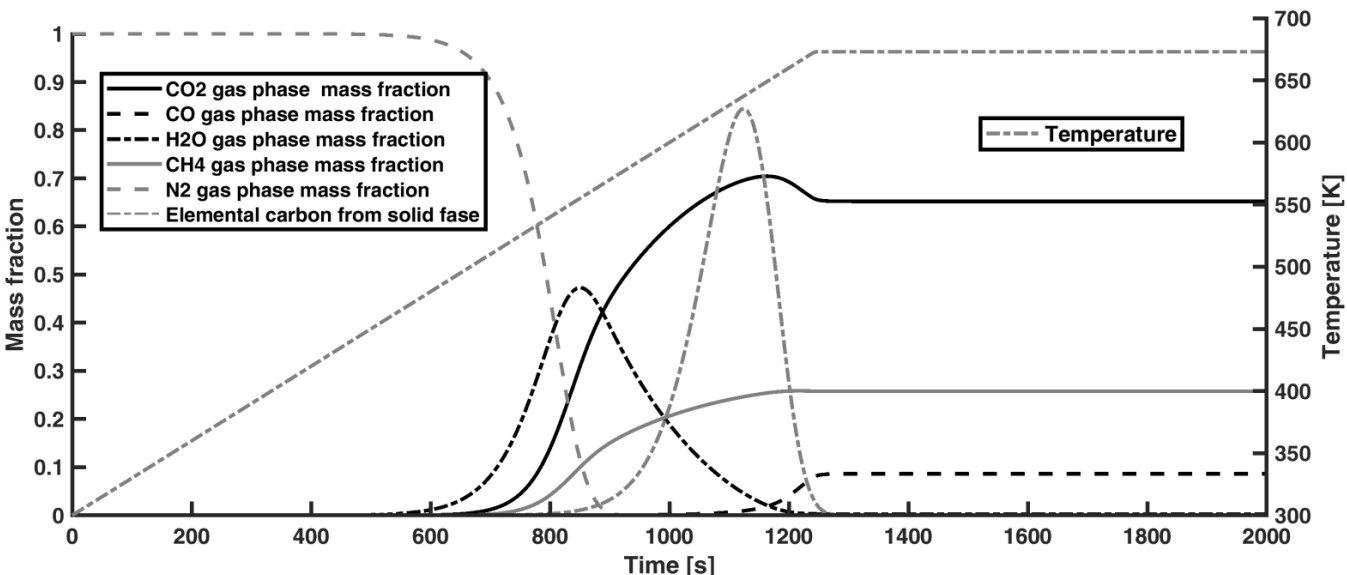

**Figure 5.** Gas phase parameters. Heating rate = 18 (K/min), $T_0$ = 300 (K), and $T_{max}$ = 673 (K).

The final mass fractions, on the global basis, were 13% and 87% for carbon and gases, respectively. The carbon mass fraction was higher than that reported in the literature [3], which stated the char from the cellulose was around 5% on a mass basis. This behaviour was due to the fact that the total carbon produced by R2 and R4 was assigned to the solid phase, and no solid carbon was present in the gas phase. Char particles produced by R2 and R4 were considered, in this work, to belong exclusively to the solid matrix, and no conversion afterward was considered (no secondary cracking reactions). Therefore, the result was a virtually very clean gas.

In actual pyrolysis processes, the gas is not clean, and an additional process is needed to separate char from the gases [1,4]. On the other hand, the temperature is low, being between 300 and 673 K, and the heating rate is relatively slow (18 K/min). This condition favours the char yield. Lower yields of char are obtained with high heating rates and higher temperatures [1,3,4].

The gas phase in this work was considered to be in chemical equilibrium permanently at one atmosphere and with a variable instead of controlled temperature. The species produced from the solid phase, except the char and activated cellulose, were equilibrated in the zero-dimensional gas system at a constant reactor pressure and current temperature conditions. In the earlier stage, the only species present in the gas phase was $N_2$ (Figure 5). As shown in Figure 6, the $N_2$ density decreased as the temperature increased at a constant pressure and volume. The $N_2$ density evolution has two stages. The first stage is caused only by the higher temperature, and then the density is lowered (in order to maintain a constant pressure). This effect causes an outward flow of pure nitrogen. Later, when practically solely $H_2O$ is released in the gas phase, the $N_2$ density decreases more rapidly due to the higher temperature and due to the displacement effect caused by the higher $H_2O$ release.

The $CO_2$ in the gas phase started to rise after the $H_2O$ species, as can be seen in Figure 5. In Figure 7 the elemental carbon, oxygen, hydrogen, and activated cellulose relative to the solid phase density are shown at a convenient scale. There is clearly correlation between the elemental species and activated cellulose.

The main elemental species produced was oxygen, and this species is the element with the higher mass fraction in the $CO_2$ molecule (i.e., 28% C and 72% O). Even though the elemental carbon released was low (between 600 and 800 s; see Figure 7), the $CO_2$ mass fraction was higher due to the oxygen content (see Figure 5).

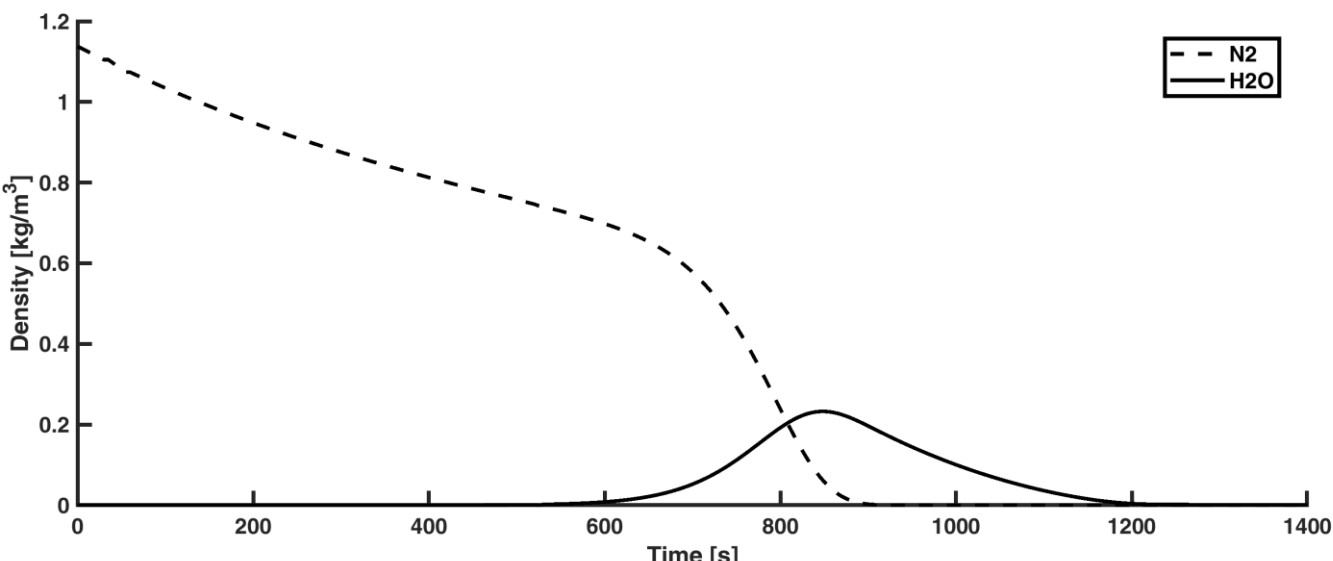

**Figure 6.** Evolution of nitrogen and water. Parameters: heating rate = 18 (K/min), $T_0$ = 300 (K), and $T_{max}$ = 673 (K).

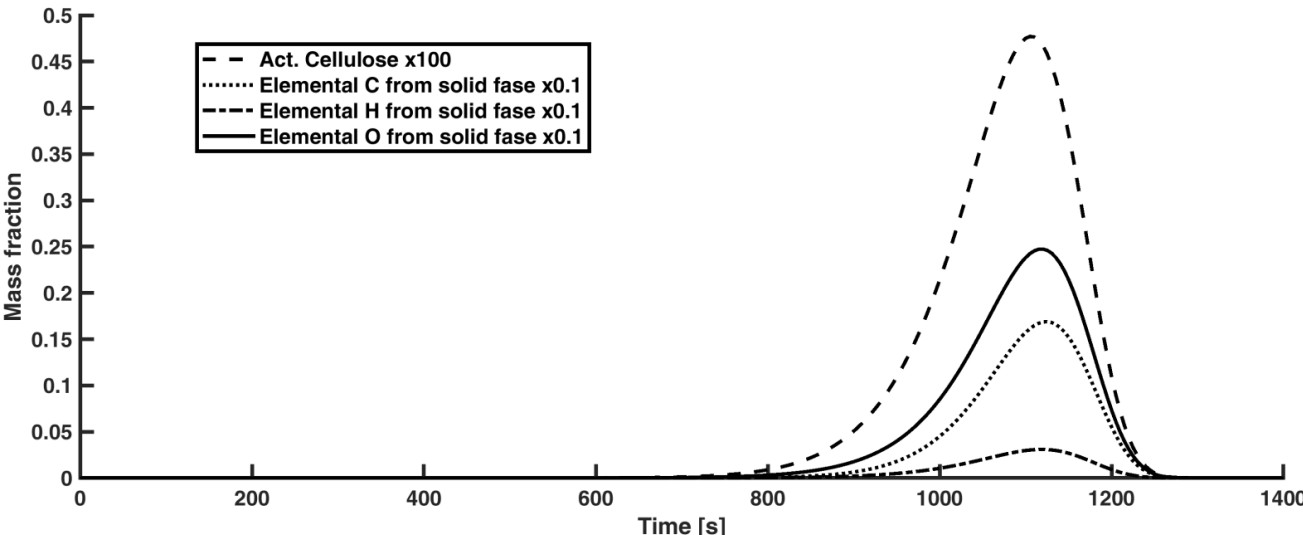

**Figure 7.** Evolution of activated cellulose, elemental carbon, hydrogen, and oxygen. Parameters: heating rate = 18 (K/min), $T_0$ = 300 (K), and $T_{max}$ = 673 (K).

The devolatilisation was practically finished at 1250 s, when the temperature was 673 K and constant. At this stage, the CO was significatively high but only due to the chemical equilibrium (Figure 5). It is also possible to observe in Figure 5 that the CO yield increased at the expense of $CO_2$ gas instead of $CH_4$.

At this stage, the species CO and $CH_4$ contributed to the SynGas heating value (HV). The gas phase composition before 1250 s is not important because it corresponds to a reminder gas inside the reactor. Therefore, if the temperature in the reactor changes, the composition will change due to the chemical equilibrium instead of devolatilisation because no more gas is produced.

Figure 8 shows the effect of the competitive characteristic of cellulose pyrolysis as mentioned in the literature [3]. For a lower rate (i.e., 1.8 K/min), the char yield was higher than the yield at 9.0 and 18.0 K/min. This was due to the solid phase remaining for a longer time at a lower temperature, which favoured the char and dehydration reactions. Considering that the lower heating rate of 1.8 K/min is 5 times lower than the 9.0 K/min heating rate, and 18.0 K/min is only 2 times higher than 9.0 K/min, as can be seen in

Figure 8, the change in the char yield from 1.8 to 9.0 K/min was higher than the change in yield of char from 9.0 to 18.0 K/min. This effect, as expected, was because of the exponential characteristic of the rate equations.

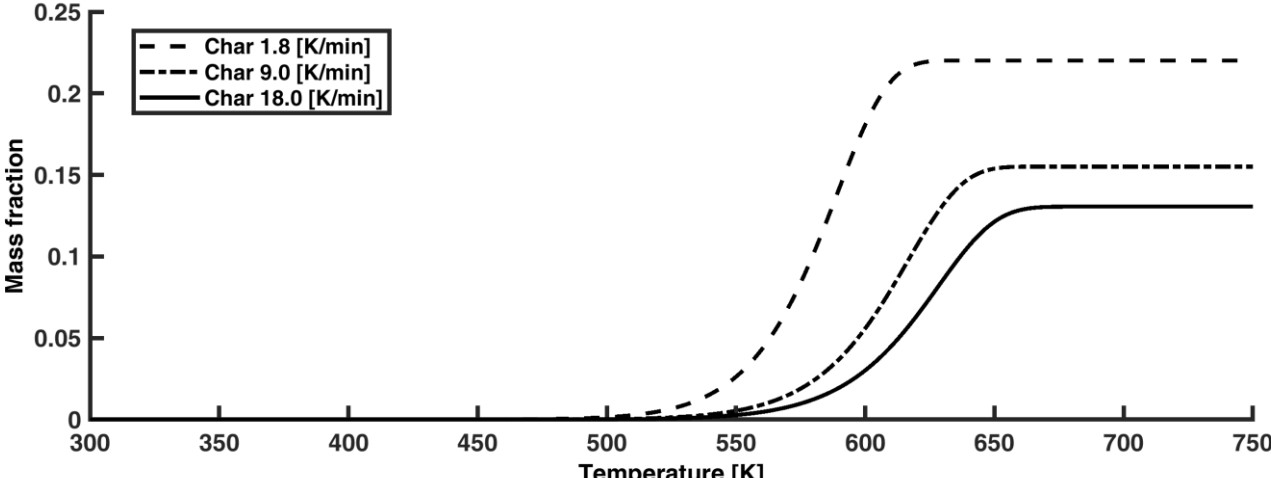

**Figure 8.** Char yield at three heating rates.

Figure 9 shows a higher peak concentration of the activated cellulose as the heating rate increased. This behaviour can explain the increase in the amount of gas released from the solid phase as the heating rate increased. The rate equation (Equation (7)) is composed of the kinetic and mass action law parts [42]. Therefore, on one hand, the higher temperature caused by the fast heating rate favours the gas yield via the kinetic part, and on the other hand, the higher concentration of activated cellulose produces a higher gas yield due to the mass action law.

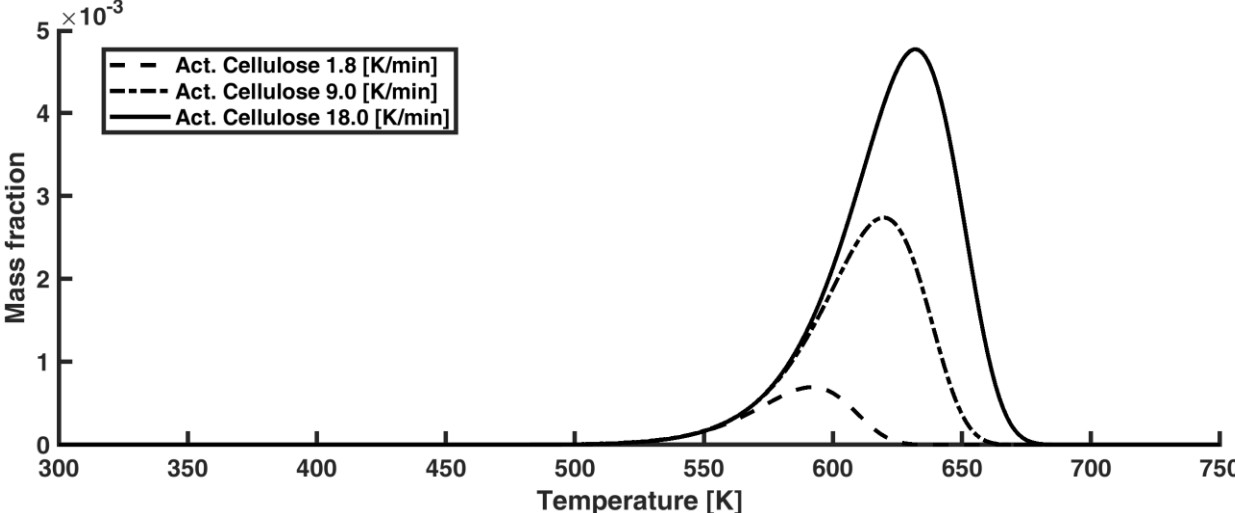

**Figure 9.** Total Acell yield at three heating rates.

Another characteristic that is possible to observe in Figure 9 is that the activated cellulose production always started at nearly the same temperature, but the temperature range in which the activated concentration was significatively high was wider as the heating rate was higher. The contrary width is seen when the mass fraction of the activated cellulose is plotted against the time (Figure 10). As can be seen in the latter figure, as the heating rate increased, the period of time in which the activated cellulose was significatively high narrowed. This effect is due to the fact that as the temperature increases, the consumption of activated cellulose to produce the species through Equations (3) and (4) increases too,

and therefore, the concentration of activated cellulose is reduced faster than at a lower heating rate.

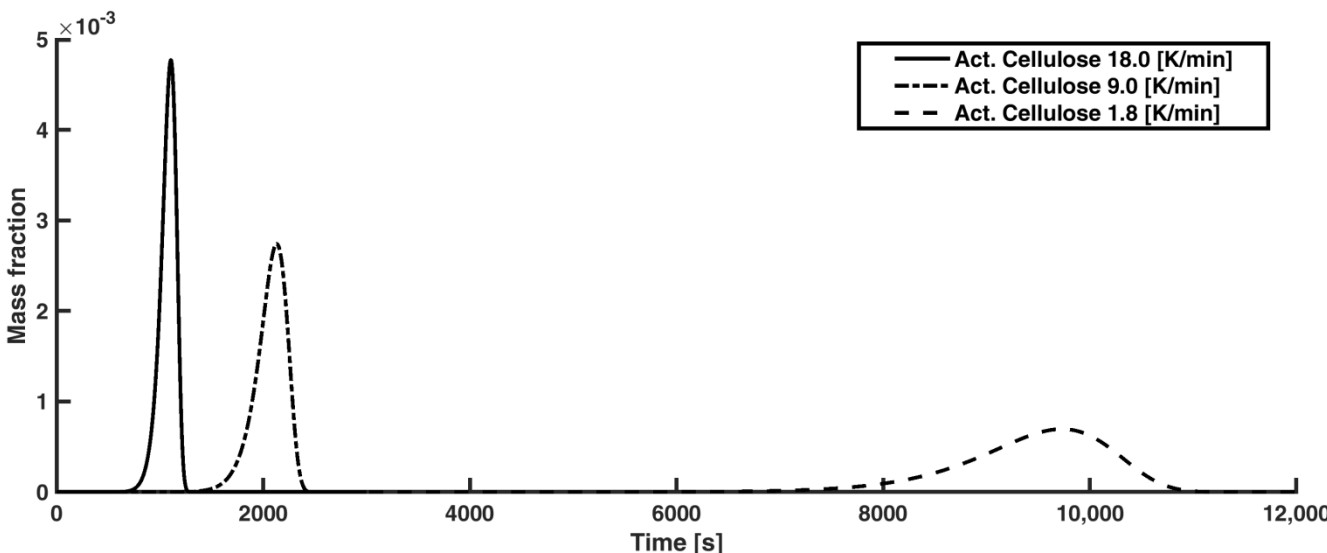

**Figure 10.** ACell evolution at three heating rates.

Finally, Figure 11 shows the extent of variability of the devolatilisation process according to the heating rate intensity [3] provided by the use of an competitive reaction mechanism [13], which is an expected behaviour. The latter implies that the higher the heating rate, the shorter the devolatilisation time. As the heating rate increased, the weight loss increased in severity, producing more gases and less char. This effect is due to the competitive characteristic of the mechanism (Table 1). Moreover, as can be observed in Table 1, the activation energy of R3 was very low. Therefore, kinetically, the reaction was active at low temperatures and faster at higher temperatures. However, it was dependent on the concentration of activated cellulose. The reaction for the activated cellulose had an activation energy higher than R1 and therefore was a bottleneck, and only at a high temperature could it compete with R1 to produce higher amounts of LVG. This dynamic and temperature-dependent characteristic is capable of reproducing the competitive characteristic in the cellulose pyrolysis process and is the reason why this model is used for simulations of thermal decomposition processes.

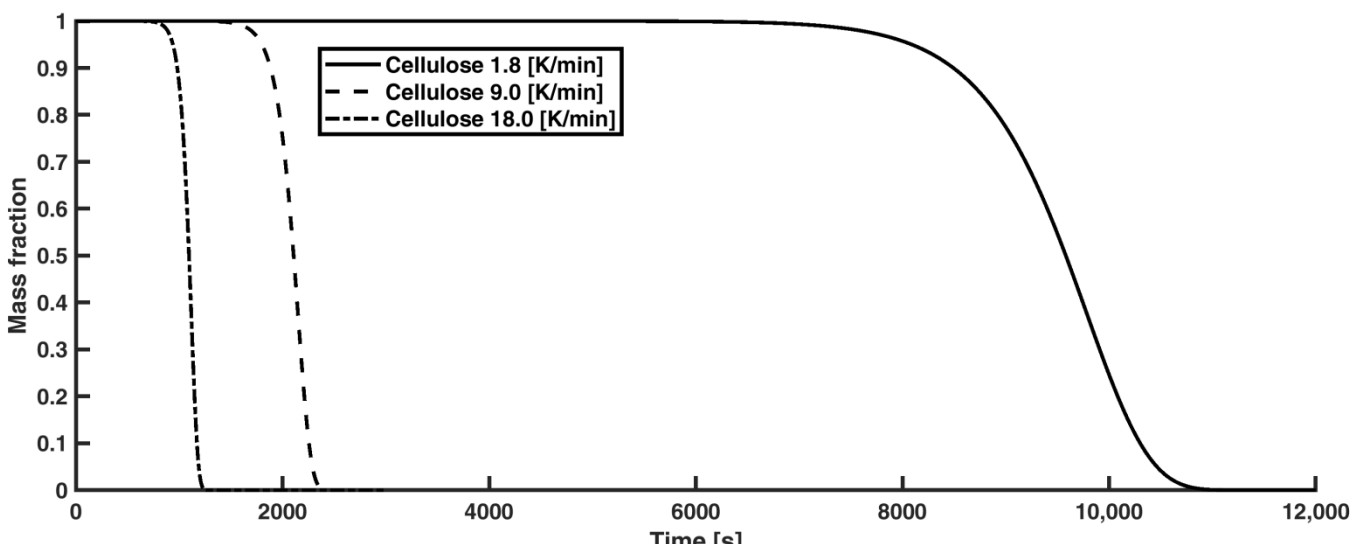

**Figure 11.** Cell reduction at three heating rates.

Figure 12a shows the experimental result obtained by Shen et al. [43] for the char residual and the rate of devolatilization from cellulose, considering a heating rate of 5 K/min in a nitrogen atmosphere. On the other hand, Figure 12b shows the final char produced and the devolatilization rate for cellulose obtained by the mechanism used in this work. By comparing both graphs, it can be seen that the maximum char production occurred around 350 °C and at 338 °C in the case of Shen et al. [43] and this work, respectively. In addition, it is also possible to observe that the maximum devolatilization rate was slightly lower for the case of the kinetic model used in this work, which reached1.71%/°C, while it was close to 2.2%/°C in the case of Shen et al. [43]. Evidently, the good fit observed here was due to the kinetic model of Ranzi et al. [13]. However, it reveals the ability of the formulation to predict the rate of kinetically controlled char formation, which is the objective of this work, while providing an approximate composition for the gas phase.

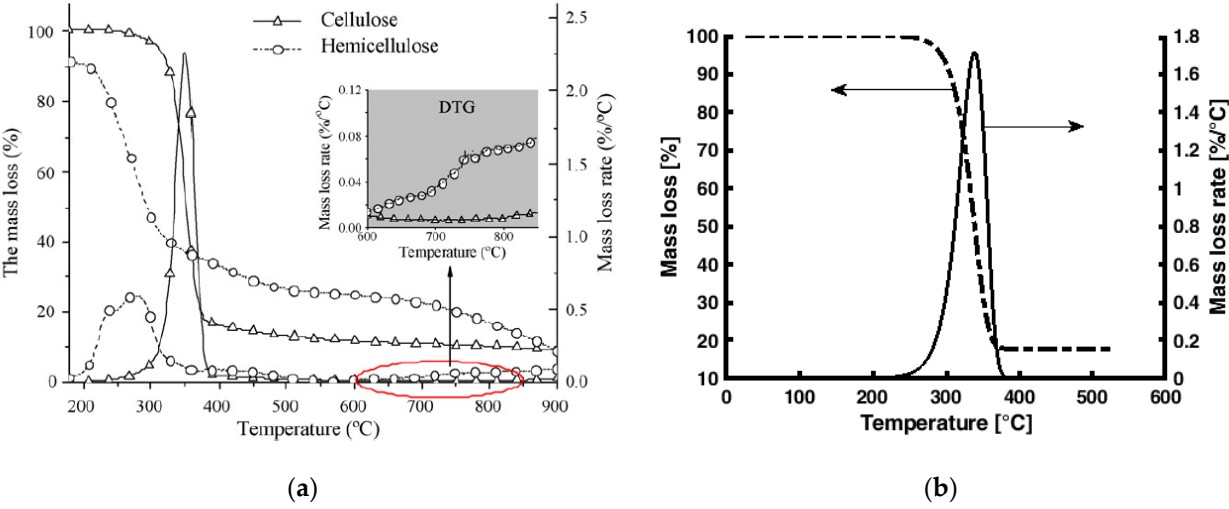

(**a**)                                                                      (**b**)

**Figure 12.** (**a**) Devolatilisationprofile for cellulose with a heating rate of 5 K/min [43]. (**b**) Devolatilisationprofile for cellulose with a heating rate of 5 K/min with the kinetic model used in this work.

Table 9 shows the distribution of the gas mass fraction in the solid phase under chemical equilibrium conditions, while Figure A1a shows the equilibrium composition for the cellulose considered in this work. Observation of the data in Table 7 and Figure A1a shows that the gas distribution showed the expected trend according to temperature; that is, the production of $CO_2$, $H_2O$, and $CH_4$ was favored at low temperatures, while it favored CO and $H_2$ at higher temperatures. This condition was also observed for the case of biomass in general, according to Lee et al. [19] and Baratieri [21] for waste material of palm and pine sawdust.

**Table 9.** Final yield of Syngas for isothermal conditions.

| | Temperature (K) | | |
|---|---|---|---|
| **Gas Phase Mass Fraction** | **573** | **773** | **873** |
| $CO_2$ | 0.250 | 0.170 | 0.770 |
| CO | 0.001 | 0.140 | 0.160 |
| $H_2O$ | 0.170 | 0.010 | 0.005 |
| $CH_4$ | 0.080 | 0.090 | 0.050 |
| $H_2$ | 0.001 | 0.001 | 0.005 |

## 5. Conclusions

As stated in the objectives in the introductory part of this article, the main goal of this work was to develop a formulation of the thermal decomposition behaviour of cellulose

pyrolysis, with a focus on the char evolution and yield and under the premise of a process in chemical pseudo-equilibrium.

From the revision of the literature, it was possible to observe that the simplest approach to studying the pyrolysis process was the equilibrium approach, but it is mainly applied to the gasification process. Furthermore, the most important parameter of this work, the char yield, cannot be obtained from the pure equilibrium models. From the point of view of the char evolution characteristics, the approach proposed here, using a parallel semi-global reaction mechanism, provides the desired results similar to those obtained by other authors.

The model's solution was consistent, and the set of ordinary differential equations was solved with minimum complexity using the totally implicit Euler scheme (TIE). The procedure to predict the gas phase composition, avoiding the kinetics and a set of homogeneous reactions, shows a significant complexity reduction. In addition, the algorithm solution does not contain any iterative loop (with the exception the internal loop in CANTERA) that always demands more computing capabilities, time, and resources. The gas phase prediction was similar to the gas compositions observed in the literature, showing that $CO_2$, $H_2O$, and $CH_4$ are mainly produced at low-temperature processes and CO at higher temperatures. In this work, $H_2$ had practically no presence due to the low temperature.

The complexity reduction was enhanced with the use of the free distributable software package CANTERA, which computed the more difficult task: the chemical equilibrium.

The aspects analysed in the final part of this article show that the model can maintain the expected behaviour for dynamical pyrolysis compared with the results shown by other authors (for the rate of formation and yield of char). The competitive factor between the gas released and char conversion as a function of the heating rate and maximum temperature especially was conserved. Furthermore, the model includes the time characteristic by definition, a factor that is not common, in its equilibrium computations.

In order to obtain the complete thermal behaviour of biomass, the extension, using the superimposed hypothesis [8], is rather simple, considering that the structure of the semi-global reaction mechanisms for hemicellulose and lignin is similar to the scheme used for cellulose.

The disadvantages of the model are that is not possible to predict intermediate gases due to the lack of kinetics for the gas phase. Additionally, the secondary cracking reactions of vapor species such as LVG, HAA, HMFU, Acetone, Acetaldehyde, or Glyoxan are not accounted for due to these species not belonging to any phase defined in this work.

The next stage of this work is to include the space in the model to account for the transport processes: advection, conduction, and radiation. This initial work provides the necessary chemical framework in order to assess a reactor and the influences on the char conversion.

The procedure to adapt an existing reaction mechanism (dynamical approach) obtained from the literature is straightforward, and it is also possible to add this procedure with a short piece of code in a computational algorithm. Therefore, in order to evaluate a pyrolysis process inside a reactor environment, reduce the developing time, and not have the complexities of the fully dynamical description, a good estimate of the pyrolysis behaviour can be obtained from a TG analysis or other reaction mechanism.

**Funding:** This research was funded by Universidad de La Frontera. The APC was funded by Universidad de La Frontera.

**Institutional Review Board Statement:** Not applicable.

**Acknowledgments:** The authors acknowledge DIUFRO project DI12-0057 "Estudio Técnico-Económico para la optimización de la producción de pellets combustibles de desechos biomásicos forestales y agrícolas, y utilización de glicerol como agente densificador energético y plastificante".

**Conflicts of Interest:** The authors declare no conflict of interest.

**Abbreviations**

Symbols

| | |
|---|---|
| $A_0$ | Pre-exponential factor (mol/(s)) |
| $[B]$ | Molar concentration |
| C++ | Computational language |
| $E$ | Activation energy (J) |
| $g$ | Volumetric fraction of gas phase |
| $k$ | Kinetic constant |
| $M$ | Molar mass (kg/(mol)) |
| $q$ | Stoichiometric matrix |
| $R$ | Universal gas constant (J/(mol K)) |
| $\dot{S}$ | Mass source rate (kg/s) |
| $S$ | Mass (kg) |
| $s$ | Volumetric fraction of solid phase |
| $T$ | Temperature (K or °C) |
| $t$ | Time |
| $X$ | Instantaneous molar fraction (mol) |
| $Y$ | Mass fraction |

Greek letters

| | |
|---|---|
| $\nu$ | Stoichiometric coefficient |
| $\rho$ | Density (kg/m$^3$) |
| $\dot{\omega}$ | Molar mass rate (mol/(s)) |

Subscripts

| | |
|---|---|
| *1,2,3,4* | Reaction number of mechanism |
| *ACell* | Activated cellulose |
| *B* | B species B in a reaction |
| *b* | Backward reaction |
| *Cell* | Cellulose |
| *Char* | Char or charcoal |
| *Devo* | Devolatilisation |
| *El* | Element species |
| *f* | Forward reaction |
| *g* | Gaseous phase |
| *i* | Species number |
| *j* | Species number |
| *n* | Current time step |
| *n+1* | Next time step |
| *s* | Solid phase |

Superscripts

| | |
|---|---|
| ′ | (Prime) denotes unbalanced quantity |
| *n* | Temperature power for Arrhenius equation |

**Appendix A**

**Figure A1.** Syngas composition for cellulose obtained in equilibrium conditions at 1 atm, considering carbon C (graphite) as a solid phase. Calculation was made with equilibrium solver of CANTERA considering the molecular formula $CH_{1.66}O_{0.83}$ for cellulose obtained from Mellin et al. [44]: $C_6H_{10}O_5$ (**a**) in mass fraction and (**b**) in molar fraction.

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
