# Peer review of "Finite Rate Reaction Mechanism Adapted for Modelling Pseudo-Equilibrium Pyrolysis of Cellulose"

_processes, doi:10.3390/pr10102131_

Round 1

Reviewer 1 Report

This kind of calculation is interesting, because it is a bridge between equilibrium-only simulations and kinetic-only ones.

My main concern is the limited literature. Please, try to expand it.

abstract
line 11:        "sense" revise wording
1. introduction - 2. pyrolysis
merge into one chapter, and expand.
3. models
this part should be expanded, and also presented with some tables and/or graphs. Also further references could be added.
lines 138-147:    are altogether redundant and ought to be eliminated.
lines 152-153:    "thermodynamic" should be replaced by "equilibrium".
lines 171-187:    references to the software documentation are needed.
lines 191-253:    this parte deserves two or three more references, but should be shorter (i.e. the referenced works should be explained more briefly).
4. description
line 307:    "stereo" (?
specify how the stoichiometrich coefficients in figure 3 and table 1 are given (moles, mass, etc).
lines 346-349:    explain better the meaning of the sentence "... same total enthalpy". does it mean that this step is adiabatic?
lines 370 onward:    symbols must be fixed, e.g.: drho/dt etc.
table 2 headings must be fixed
lines 456-462:        it is not clear why the overall gas density rho_g must be recalculated, if volume, pressure and temperature are fixed as boundary conditions. Please, explain this point more clearly. probably it would also be better to move fig 4 above.
In general, an equation for the temperature is missing. Otherwise, it should be specified if it is imposed.
5. Results - 6. Conclusions
The graphs would be clearer if also symbols were added, besid differently dotted lines.
The author state that their results are in line with lit. data for the gas, but don't produce any graph or table to support this statement. An extension in this sense is strongly recommended.

Author Response

Response to Reviewer 1 Comments

Dear reviewer, first of all I appreciate the time spent by you to review this manuscript and give it the opportunity to be published. Below I answer all the 17 comments made one by one. I comment that in addition to the observations made by you, observations have also been made for another reviewer. Consequently the text has changed and therefore some observations suggested by you are not applicable.

Comment 1

My main concern is the limited literature. Please, try to expand it. Answer: Thanks for your observation. Added new 17 refences.

Comment 2

abstract
line 11:        "sense" revise wording. Answer: Thanks for your observation. Changed by “approach”

Comment 3
1. introduction - 2. pyrolysis
merge into one chapter, and expand. Answer: Thanks for your observation. Merged and expanded.

Comment 4
3. models
this part should be expanded, and also presented with some tables and/or graphs. Also further references could be added. Answer: Thanks for your observation. The section was expanded and 17 new references were added.

Comment 5
lines 138-147:    are altogether redundant and ought to be eliminated. Answer: Thanks for your observation. Agree, removed. Only maintined “Thermodynamics is a powerful discipline that allows predicting thermal processes behavior.”

Comment 6
lines 152-153:    "thermodynamic" should be replaced by "equilibrium". Answer: Thanks for your observation. Agree and modified.

Comment 7
lines 171-187:    references to the software documentation are needed. Answer: Thanks for your observation. Reference added “Goodwin, D.G., 2001. Cantera User’s Guide Fortran Version Release 1.2.”

Comment 8
lines 191-253:    this parte deserves two or three more references, but should be shorter (i.e. the referenced works should be explained more briefly). Answer: Thanks for your observation. 17 new related references were added and the description of Baratieri, Altafini, Lee and Gobel articles reduced.

Comment 9
4. description
line 307:    "stereo" (?.Answer: Thanks for your observation. Following modification is inserted in text. The later species have the same elemental composition of cellulose but with different and properties such as formation enthalpy [12]. “

Comment 10

specify how the stoichiometrich coefficients in figure 3 and table 1 are given (moles, mass, etc). Answer: Thanks for your observation. Following modification is inserted in text. Both reactions have their own kinetic constants shown in Table 1 in where  is the pre-exponential factor,  the activation energy and T (-) the temperature factor.”

Comment 11
lines 346-349:    explain better the meaning of the sentence "... same total enthalpy". does it mean that this step is adiabatic? Answer: Thanks for your observation. Yes. This stage is considered as adiabatic. Fixed by explanation in text.

Comment 12
lines 370 onward:    symbols must be fixed, e.g.: drho/dt etc. Answer: Thanks for your observation. Fixed.

Comment 13
table 2 headings must be fixed. Answer: Thanks for your observation. Fixed.

Comment 14

lines 456-462:  it is not clear why the overall gas density rho_g must be recalculated, if volume, pressure and temperature are fixed as boundary conditions. Please, explain this point more clearly. probably it would also be better to move fig 4 above. Answer: Thanks for your observation. Fixed. The density must be recalculated because of the volume of the solid phase is different from that of the gas phase. The balance factor is s/g i.e. volumetric fractions of solid and gas phase respectively and defined in the nomenclature.

Comment 15

In general, an equation for the temperature is missing. Otherwise, it should be specified if it is imposed.

Answer: Thanks for your observation. The equation for temperature is not missing. For the current solution process the temperature is an arbitrary parameter that is pre-defined. Specifically, it is defined as a temperature ramp of: 1.8, 9.0 and 18.0 K/min. Therefore, energy equation is not necessary to determine the temperature. However, if this formulation is applied to a CFD simulation for example, then the temperature will be a local variable depending on the corresponding energy transport equation. The table 8 in the manuscript defines the temperature ramp. This has been formally made explicit in line ___ of the new text.

Comment 16
5. Results - 6. Conclusions
The graphs would be clearer if also symbols were added, besid differently dotted lines.

Answer: Thanks for your observation. New figures with graphics are produced with improved legend.

Comment 17
The author state that their results are in line with literature data for the gas, but don't produce any graph or table to support this statement. An extension in this sense is strongly recommended.

Answer: Thanks for your observation. The following part was added in order to provide a comparison. “Figure 13a shows the experimental result obtained by Shen et al. [42] for char residual and the rate of devolatilization from cellulose considering a heating rate of 5 K/min in nitrogen atmosphere. On the other hand, Figure 13b shows the final char produced and the devolatilization rate for cellulose obtained by the mechanism used in this work. Comparing both graphs, it can be seen that the maximum char production occurs around 350°C and at 338°C in the case of Chen et al. [42] and this work respectively. In addition, it is also possible to observe that the maximum devolatilization rate is slightly lower for the case of the kinetic model used in this work, which reaches 1.71 %/°C while it is close to 2.2 %/°C in the case of Shen et al. [42]. Evidently the good fit observed here is due to the kinetic model of Ranzi et al. [13]. However, it reveals the ability of the formulation to predict the rate of kinetically controlled char formation, which is the objective of this work while providing an approximate composition for the gas phase.

Also it was added a verification for gas phase as follow: “Table 7 shows the distribution of the gas mass fraction in the solid phase under chemical equilibrium conditions while Figure 14 a) shows the equilibrium composition for the cellulose considered in this work. Observation of the data in Table 7 and Figure 14 a) show that the gas distribution shows the expected trend according to temperature, that is, the production of CO2, H2O and CH4 is favored at low temperature while they are favored CO and H2 at higher temperatures. This condition is also observed for the case of biomass in general, as can be seen in Figures 15 and 16 of the Appendix I for waste material of Palm and Pine sawdust.“

Table 7. Final yield of Syngas for isothermal conditions.

Temperature K

Gas phase Mass fraction

573

773

873

CO2

0.250

0.170

0.770

CO

0.001

0.140

0.160

H2O

0.170

0.010

0.005

CH4

0.080

0.090

0.050

H2

0.001

0.001

0.005

Reviewer 2 Report

The paper deals with the Finite rate reaction mechanism adapted for modelling pseudo - equilibrium pyrolysis of cellulose.
The paper, in the opinion of the reviewer, is worth publishing
at processes Journal.
While the authors have made considerable research effort,
the presentation of the paper and the results must be improved.
The reviewer suggested corrections to the authors,
which of accepted and inserted in the paper,
then the paper can be accepted for publication
in the journal.

Comment 1
The author must format the paper according to the journal's instructions
Line 7
Abstract:
Line 23
Keywords:

Comment 2
Exteded text editing
Line 34
The main objective    is to obtain,
The author should replace (delete the extra space)
The main objective is to obtain,

Line 62
phase can undergo further   heterogeneous
The author should replace (delete the extra space)
phase can undergo further heterogeneous

Comment 3
Line 40
Line 278
It's not so good to start the sub-section at the bottom of the page without using text.

Comment 4
Line 47
Basu [2] and Bridgwater [4]
The author must replace (There is not the name of the author [2] - Always if there are two or more authors the name of the first author and et al.)
Wurzenberger [2] and Bridgwater et al. [4]

Line 66
given by Diebold [6],
The author must replace
given by Diebold et al. [6],

Line 91
point of view, [1], [3], [4], [7].
The author must replace
point of view, [1, 3, 4, 7].

Line 113
Blasi [3] and Branca [9]
The author must replace
Blasi [3] and Branca et al. [9]

Line 126
and more char [1], [3].
The author must replace
and more char [1, 3].

Comment 5
According to the journal's instructions
References: References must be numbered in order of appearance in the text
(including table captions and figure legends) and listed individually at the end of the manuscript.

Line 122: Ref. [10]
Line 128: Ref. [21]
From [10] to [21]

Line 158
From [13] to [22]

The author must format (renumber) the paper according to the journal's instructions.

Comment 6
The author should replace
Fig. to Figure

Comment 7
Line 133
given by Diebold [11] and
There is no Diebold at [11].
The author must check the names of the Ref. authors.

given by Ahmed et al. [11] and

Line 134
mechanism given by Ranzi [12].
The author should replace
mechanism given by Ranzi et al. [12].

Comment 8
Line 164
As stated by Vonka [15]
The author should replace
As stated by Vonka et al. [15]

Comment 9
Lines 178 - 179
(See http://www.cantera.org).

The author must use ref. number for this.

Line 188
an equilibrium state [16].    Also in an entrained
The author should replace (delete the extra space)
an equilibrium state [16]. Also in an entrained

Line 192
Baratieri [18] used
The author should replace
Baratieri et al. [18] used

Line 201 + Line 204 + Line 340
The author must delete the extra spaces.

Line 205
Altafini [17] has performed
The author should replace
Altafini et al. [17] has performed

Line 222
Gobel [19] developed
The author should replace
Gobel et al. [19] developed

Line 242
was developed by Lee [16].
The author should replace
was developed by Lee et al. [16].

Comment 10
Figure 2
CONTROLED
The author must check the paper for typographical and spelling errors.

Comment 11
Lines 295, 298 and 304
by Ranzi [12]
The author should replace
by Ranzi et al. [12]

Line 300
by Miller [8]
The author should replace
by Miller et al. [8]

Comment 12
According to the journal's instructions
All Figures, Schemes and Tables should be inserted into the main text close
to their first citation and must be numbered following their number of appearance (Figure 1, Scheme I, Figure 2, Scheme II, Table 1, etc.).
Tables should be placed in the main text near to the first time they are cited.
The author must move the Table 1 into the main text close to their first citation.

Comment 13
Line 367
From [19] to [23]
The author must format (renumber) the paper according to the journal's instructions.

Comment 14
The numbering of the equations should be at the end of the line on the right side of the text.

Line 376
Why R5 and no 5?

Line 379
Since Ranzi [12]
The author should replace
Since Ranzi et al. [12]

Comment 15
From equation (24) to (26).
25 is missing.

Line 465
cellulose (eq. 14), activated cellulose (15) and char (15);
The author must check if the numbers of the equations are right: char (16?)

Line 476
(see Table)
The author must define the number of the Table.

There are 2 equations (31).
The author must renumber.

Comment 16
Figure 4
Update all varaibles and ....
The author must check the paper for typographical and spelling errors.

Comment 17
Line 544
1,9, 8,0 and 18,0

The author must define the numbers with a better way.

Comment 18
The author must move the Table 8 into the main text close to their first citation.

Comment 19
The author must increase the visibility of the Figures 5, 6, 7, 8, 9, 10
For Figure 7: there are two curves, but at the Figure's title  Grey: (-) H2O, (--). Black:(-) N2?
Something is missing. The author must check.

Comment 20
Line 606
from gases [1], [4].
The author should replace
from gases [1, 4].

Line 609
[1], [3] and [4].
The author should replace
[1, 3, 4].

Comment 21
Line 680
the premise the premise
The author must check the paper for typographical and spelling errors.

Comment 22
The author must format the paper according to the journal's instructions.

Comment 23
References
Increase the number of the reference papers including (primarily) from MDPI Journals
The authors use 0 papers from Processes journal / 0 MDPI Journals / 23 papers from journals (References)
Τhe number for papers from MDPI journals
is considered insufficient (in reviewer's opinion).

Comment 24
The authors must format the References according to the journal's instructions.
References should be described as follows, depending on the type of work:
Journal Articles:
1. Author 1, A.B.; Author 2, C.D. Title of the article. Abbreviated Journal Name Year, Volume, page range.

The author delete the [ ], " and must check if all data is in the Ref.
For examble: Ref 16 - The name of the journal is missing (Fuel)

Author Response

Response to Reviewer 2 Comments

Dear reviewer, first of all I appreciate the time spent by you to review this manuscript and give it the opportunity to be published. Below I answer all the 24 comments made one by one. I comment that in addition to the observations made by you, observations have also been made for another reviewer. Consequently the text has changed and therefore some observations suggested by you are not applicable.

Comment 1
The author must format the paper according to the journal's instructions
Line 7
Abstract:
Line 23
Keywords:
Answer: Fixed

Comment 2
Exteded text editing
Line 34
The main objective    is to obtain,
The author should replace (delete the extra space)
The main objective is to obtain,

Answer: Fixed

Line 62
phase can undergo further   heterogeneous
The author should replace (delete the extra space)
phase can undergo further heterogeneous

Answer: Fixed

Comment 3
Line 40
Line 278
It's not so good to start the sub-section at the bottom of the page without using text.

Answer: Fixed

Comment 4
Line 47
Basu [2] and Bridgwater [4]
The author must replace (There is not the name of the author [2] - Always if there are two or more authors the name of the first author and et al.)
Wurzenberger [2] and Bridgwater et al. [4]

Answer: Fixed

Line 66
given by Diebold [6],
The author must replace
given by Diebold et al. [6],

Answer: Fixed

Line 91
point of view, [1], [3], [4], [7].
The author must replace
point of view, [1, 3, 4, 7].

Answer: Fixed

Line 113
Blasi [3] and Branca [9]
The author must replace
Blasi [3] and Branca et al. [9]

Answer: Fixed

Line 126
and more char [1], [3].
The author must replace
and more char [1, 3].

Answer: Fixed

Comment 5
According to the journal's instructions
References: References must be numbered in order of appearance in the text
(including table captions and figure legends) and listed individually at the end of the manuscript. Answer: Additional references were added and the entire ref are renumbered.

Line 122: Ref. [10]
Line 128: Ref. [21]
From [10] to [21]

Line 158
From [13] to [22]

The author must format (renumber) the paper according to the journal's instructions.

Answer: Fixed

Comment 6
The author should replace
Fig. to Figure

Answer: Fixed

Comment 7
Line 133
given by Diebold [11] and
There is no Diebold at [11].
The author must check the names of the Ref. authors.

given by Ahmed et al. [11] and

Answer: Fixed

Line 134
mechanism given by Ranzi [12].
The author should replace
mechanism given by Ranzi et al. [12].

Answer: Fixed

Comment 8
Line 164
As stated by Vonka [15]
The author should replace
As stated by Vonka et al. [15]

Answer: Fixed

Comment 9
Lines 178 - 179
(See http://www.cantera.org).

The author must use ref. number for this.

Answer: New ref for CANTERA web page is added and other ref for user guide documentation.

Line 188
an equilibrium state [16].    Also in an entrained
The author should replace (delete the extra space)
an equilibrium state [16]. Also in an entrained

Answer: Fixed

Line 192
Baratieri [18] used
The author should replace
Baratieri et al. [18] used

Answer: Fixed

Line 201 + Line 204 + Line 340
The author must delete the extra spaces.

Answer: Fixed

Line 205
Altafini [17] has performed
The author should replace
Altafini et al. [17] has performed

Answer: Now the author is not referenced by their name

Line 222
Gobel [19] developed
The author should replace
Gobel et al. [19] developed

Answer: Fixed

Line 242
was developed by Lee [16].
The author should replace
was developed by Lee et al. [16].

Answer: Fixed

Comment 10
Figure 2
CONTROLED
The author must check the paper for typographical and spelling errors.

Answer: Fixed

Comment 11
Lines 295, 298 and 304
by Ranzi [12]
The author should replace
by Ranzi et al. [12]

Answer: Fixed

Line 300
by Miller [8]
The author should replace
by Miller et al. [8]

Answer: Fixed

Comment 12
According to the journal's instructions
All Figures, Schemes and Tables should be inserted into the main text close
to their first citation and must be numbered following their number of appearance (Figure 1, Scheme I, Figure 2, Scheme II, Table 1, etc.).
Tables should be placed in the main text near to the first time they are cited.
The author must move the Table 1 into the main text close to their first citation.

Answer: Fixed

Comment 13
Line 367
From [19] to [23]
The author must format (renumber) the paper according to the journal's instructions.

Answer: Fixed

Comment 14
The numbering of the equations should be at the end of the line on the right side of the text.

Line 376
Why R5 and no 5?

Answer: All equation are renumbered

Line 379
Since Ranzi [12]
The author should replace
Since Ranzi et al. [12]
Answer: Fixed

Comment 15
From equation (24) to (26).
25 is missing.

Answer: All equation are renumbered

Line 465
cellulose (eq. 14), activated cellulose (15) and char (15);
The author must check if the numbers of the equations are right: char (16?)

Answer: Fixed

Line 476
(see Table)
The author must define the number of the Table.

Answer: Fixed

There are 2 equations (31).
The author must renumber.

Answer: All equation are renumbered

Comment 16
Figure 4
Update all varaibles and ....
The author must check the paper for typographical and spelling errors.

Answer: Fixed

Comment 17
Line 544
1,9, 8,0 and 18,0

Answer: decimal period has been change to a point.

The author must define the numbers with a better way.

Comment 18
The author must move the Table 8 into the main text close to their first citation.

Answer: Fixed

Comment 19
The author must increase the visibility of the Figures 5, 6, 7, 8, 9, 10
For Figure 7: there are two curves, but at the Figure's title  Grey: (-) H2O, (--). Black:(-) N2?
Something is missing. The author must check.

Answer: Fixed

Comment 20
Line 606
from gases [1], [4].
The author should replace
from gases [1, 4].

Answer: Fixed

Line 609
[1], [3] and [4].
The author should replace
[1, 3, 4].

Answer: Fixed

Comment 21
Line 680
the premise the premise
The author must check the paper for typographical and spelling errors.

Answer: Fixed

Comment 22

The author must format the paper according to the journal's instructions.

Answer: Fixed

Comment 23
References
Increase the number of the reference papers including (primarily) from MDPI Journals
The authors use 0 papers from Processes journal / 0 MDPI Journals / 23 papers from journals (References)
Τhe number for papers from MDPI journals
is considered insufficient (in reviewer's opinion).

Answer: Four MDPI articles were added related with the topics of the article. One from Energies a 3 from Procesess.

Comment 24
The authors must format the References according to the journal's instructions.
References should be described as follows, depending on the type of work:
Journal Articles:
1. Author 1, A.B.; Author 2, C.D. Title of the article. Abbreviated Journal Name YearVolume, page range.

Answer: Fixed

The author delete the [ ], " and must check if all data is in the Ref.
For examble: Ref 16 - The name of the journal is missing (Fuel)

Answer: Fixed

Reviewer 3 Report

Title: Finite rate reaction mechanism adapted for modeling pseudo-equilibrium pyrolysis of cellulose

Journal: Processes

Main comments: The current work represents plagiarism of previously published work by the same author entitled: “Finite Rate Reaction Mechanism Adapted for Modeling Pseudo-Equilibrium Pyrolysis of Cellulose” published in Preprints 2022, 28 Feb 2022 2022020351 (doi: 10.20944/preprints202202.0351.v1). I ask the Editor to pay attention to such gross violations of ethical norms, especially for a reputable Journal such as “Processes”. Manuscript should be rejected.

Author Response

Response to Reviewer 3 Comments

Dear reviewer, first of all I appreciate your comment and your concern about plagiarism in order to maintain the prestige of Processes journal. I have consulted the editorial office (to Codrina Solcanu) regarding plagiarism and they indicate that it does not correspond. This is due to the fact that MDPI itself suggests sending a work to PrePrint to get feedback before sending it to a journal of the MDPI itself. I still hope that you can revise the manuscript at the next stage.

Best Regards.

Round 2

Reviewer 2 Report

Comment 1 (Previous Comment 1) 

The author must format the paper according to the journal's instructions

Line 7

Abstract:

Line 23

Keywords:

Abstract:

This manuscript is related

The author must replace

Abstract: This manuscript is related

Keywords: 

Cellulose, pyrolysis, chemical equilibrium, chemical kinetics.

The author must replace

Keywords: Cellulose, pyrolysis, chemical equilibrium, chemical kinetics.

Comment 2 (Previous Comment 2) 

Exteded text editing

Line 32

The main objectiveis to obtain, 

The author must replace

The main objective is to obtain, 

Comment 3 (Previous Comment 4) 

Line 47

Basu [2] and Bridgwater [4]

The author must replace (There is not the name of the author [2] - Always if there are two or more authors the name of the first author and et al.)

Wurzenberger [2] and Bridgwater et al. [4]

Line 58

Bridgwater [4] explains

The author must replace

Bridgwater et al. [4] explains

Line 61

by Dieboldet al.[6], is the 

The author must replace

by Dieboldet al. [6], is the 

Line 98

The latter correspond to am hypothesis with a good

The author must check the paper for typographical and spelling errors.

Line 118

Shafizadeh[11].

The author must replace

Shafizadeh [11].

Line 125

The author must format the paper according to the journal's instructions

Line 127

thermal processesbehavior. 

The author must replace

thermal processes behavior. 

Line 133

equilibriumdescription

The author must replace

equilibrium description

Line 154

CANTERA[17].

The author must replace

CANTERA [17].

Line 163

state   [19]. 

The author must replace

state [19]. 

Line 165

and air to fuelratio (ER) for the reduction phase.In the literature,

The author must replace

and air to fuel ratio (ER) for the reduction phase. In the literature,

Line 166

of articlesapplying

The author must replace

of articles applying

Line 167

whereapplied

The author must replace

where applied

Line 173

Since in a actualapplication

The author must replace

Since in a actual application

Line 199

source.Examples

The author must replace

source. Examples

Line 215

in the literature[31 - 33]

The author must replace

in the literature [31 - 33]

Line 222

Gøbelet al. [34] 

The author must replace

Gøbel et al. [34] 

Line 225

transport.Kinetics

The author must replace

transport. Kinetics

Line 306

factor,E

The author must replace

factor, E

Line 478

?(see Table 1),

The author must replace

? (see Table 1),

Line 735

Acknowledgementsforthe

The author must replace

Acknowledgements for the

Line 731

from aTG analysis

The author must replace

from a TG analysis

Comment 4

Figure 2

CONTANT PRESSURE

Exteded text editing

The author must check the paper for typographical and spelling errors.

Comment 5 (previous Comment 14)

The numbering of the equations should be at the end of the line on the right side of the text.

Comment 6

The authos must check the equation 8.

Comment 7

Line 406

by Turns [14]

The author must check (Cengel?)

Comment 8

Lines 471 - 473

The equation system is solved numerically with a simple as possible method at this starting stage, 

leaving a more complex solution scheme for future works. The most simple solution is provided by 

the Euler method, however the latter has the disadvantage that it can produce mass inconsistencies.  

The author must explain with more details.

Comment 9

Figure 4

Exteded text editing

Temperatue

coeficients

The author must check the paper for typographical and spelling errors.

Comment 10

According to the journal's instructions

All Figures, Schemes and Tables should be inserted into the main text close 

to their first citation and must be numbered following their number of appearance (Figure 1, Scheme I, Figure 2, Scheme II, Table 1, etc.).

Tables should be placed in the main text near to the first time they are cited.

The author must move the Table 8 after the Table 7.

Comment 11

The author must increase the visibility of the Figures 5, 6, 7, 8, 9, 10 (like Figure 14).

Comment 12 (previous Comment 24)

The authors must format the References according to the journal's instructions.

References should be described as follows, depending on the type of work:

Journal Articles:

1. Author 1, A.B.; Author 2, C.D. Title of the article. Abbreviated Journal Name Year, Volume, page range.

Author Response

Response to Reviewer 2 Comments

Dear reviewer, first of all I appreciate the time spent by you to review this manuscript and give it the opportunity to be published. Below I answer all the 12 comments made one by one.

Comment 1 (Previous Comment 1) 

The author must format the paper according to the journal's instructions

Line 7

Abstract:

Line 23

Keywords:

Abstract:

This manuscript is related

The author must replace

Abstract: This manuscript is related

Keywords: 

Cellulose, pyrolysis, chemical equilibrium, chemical kinetics.

The author must replace

Keywords: Cellulose, pyrolysis, chemical equilibrium, chemical kinetics.

Answer: Fixed Journal format has been applied.

Comment 2 (Previous Comment 2) 

Exteded text editing

Line 32

The main objectiveis to obtain, 

The author must replace

The main objective is to obtain, 

Answer: Fixed

Comment 3 (Previous Comment 4) 

Line 47

Basu [2] and Bridgwater [4]

The author must replace (There is not the name of the author [2] - Always if there are two or more authors the name of the first author and et al.)

Wurzenberger [2] and Bridgwater et al. [4]

Line 58

Bridgwater [4] explains

The author must replace

Bridgwater et al. [4] explains

Line 61

by Dieboldet al.[6], is the 

The author must replace

by Dieboldet al. [6], is the 

Line 98

The latter correspond to am hypothesis with a good

The author must check the paper for typographical and spelling errors.

Line 118

Shafizadeh[11].

The author must replace

Shafizadeh [11].

Line 125

The author must format the paper according to the journal's instructions

Line 127

thermal processesbehavior. 

The author must replace

thermal processes behavior. 

Line 133

equilibriumdescription

The author must replace

equilibrium description

Line 154

CANTERA[17].

The author must replace

CANTERA [17].

Line 163

state   [19]. 

The author must replace

state [19]. 

Line 165

and air to fuelratio (ER) for the reduction phase.In the literature,

The author must replace

and air to fuel ratio (ER) for the reduction phase. In the literature,

Line 166

of articlesapplying

The author must replace

of articles applying

Line 167

whereapplied

The author must replace

where applied

Line 173

Since in a actualapplication

The author must replace

Since in a actual application

Line 199

source.Examples

The author must replace

source. Examples

Line 215

in the literature[31 - 33]

The author must replace

in the literature [31 - 33]

Line 222

Gøbelet al. [34] 

The author must replace

Gøbel et al. [34] 

Line 225

transport.Kinetics

The author must replace

transport. Kinetics

Line 306

factor,E

The author must replace

factor, E

Line 478

(see Table 1),

The author must replace

? (see Table 1),

Line 735

Acknowledgementsforthe

The author must replace

Acknowledgements for the

Line 731

from aTG analysis

The author must replace

from a TG analysis

Answer: All Fixed

Comment 4

Figure 2

CONTANT PRESSURE

Exteded text editing

The author must check the paper for typographical and spelling errors.

Answer: Fixed

Comment 5 (previous Comment 14)

The numbering of the equations should be at the end of the line on the right side of the text.

Answer: Fixed

Comment 6

The author must check the equation 8.

Answer: Equation 8 is the correct one for modified Arrhenius. Only the dependency of t (time) has been changed to T (Temperature).

Comment 7

Line 406

by Turns [14]

The author must check (Cengel?)

Answer: Reference renumbered by [42]

Comment 8

Lines 471 - 473

The equation system is solved numerically with a simple as possible method at this starting stage, 

leaving a more complex solution scheme for future works. The most simple solution is provided by 

the Euler method, however the latter has the disadvantage that it can produce mass inconsistencies.  

The author must explain with more details.

Answer: Fixed . Sentence “i.e. negative density” has been added.

Comment 9

Figure 4

Exteded text editing

Temperatue

Coeficients

 The author must check the paper for typographical and spelling errors.

Answer: Fixed

Comment 10

According to the journal's instructions

All Figures, Schemes and Tables should be inserted into the main text close 

to their first citation and must be numbered following their number of appearance (Figure 1, Scheme I, Figure 2, Scheme II, Table 1, etc.).

Tables should be placed in the main text near to the first time they are cited.

The author must move the Table 8 after the Table 7.

Answer: Fixed

 Comment 11

The author must increase the visibility of the Figures 5, 6, 7, 8, 9, 10 (like Figure 14).

Answer: Fixed

Comment 12 (previous Comment 24)

The authors must format the References according to the journal's instructions.

References should be described as follows, depending on the type of work:

Journal Articles:

  1. Author 1, A.B.; Author 2, C.D. Title of the article. Abbreviated Journal NameYearVolume, page range.

Answer: Fixed

Reviewer 3 Report

Accept as it is.

Author Response

Response to Reviewer 3 Comments

Dear reviewer, I appreciate the second chance you have given to this manuscript as well as your recommendation to the editor to be published in the journal Processes.

Best Regards.

Round 3

Reviewer 2 Report

Comment 1

Line 48

Basu [2] and Bridgwater et al.[4] classify

The author must replace

Wurzenberger [2] and Diebold et al. [4]

Line 182

phase.In the literature,

The author must replace

phase. In the literature,

Line 228

Apsen Plus subroutines.In all of these

The author must replace

Apsen Plus subroutines. In all of these

Line 231

experimental data.The most

The author must replace

experimental data.The most

Line 330

in Table 1in where

The author must replace

in Table 1 in where

Comment 2

Lines 342 and 819

It's not so good to start the sub-section at the bottom of the page without using text.

The author must format (move the title at the other page).

Lines 446 and 477

The Table must be accompanied on the same page as the Table's title.

The author must format (move the title at the other page).

Comment 3

Figure 7

Tiempo [s]

The author must check for spelling and typographical errors.

Comment 4

Line 826

The author must format the References according to journal's instructions (delete the gap - page 27) 

For all References (1 - 43)

1 Basu, P.;

The author must replace (add a .)

1. Basu, P.;

Author Response

Response to Reviewer 2 Comments

Dear reviewer, first of all I appreciate the time spent by you to review this manuscript and give it the opportunity to be published. Below I answer all the 4 comments made one by one.

Comment 1

Line 48

Basu [2] and Bridgwater et al.[4] classify

The author must replace

Wurzenberger [2] and Diebold et al. [4]

Answer: Actually Basu and Bridgewater states the sentence not Wurzenberger. Thank you very much for the observation. The new citation is “Authors like, Basu [1] and Diebold et al. [4] classify…”

Line 182

phase.In the literature,

The author must replace

phase. In the literature,

Line 228

Apsen Plus subroutines.In all of these

The author must replace

Apsen Plus subroutines. In all of these

Line 231

experimental data.The most

The author must replace

experimental data.The most

Line 330

in Table 1in where

The author must replace

in Table 1 in where

Answer: This time space issues was totally checked. Apparently it was a problem with the word processor.

Comment 2

Lines 342 and 819

It's not so good to start the sub-section at the bottom of the page without using text.

The author must format (move the title at the other page).

Answer: Text, Tables and Figures were relocated taking care about the closeness between them.

Lines 446 and 477

The Table must be accompanied on the same page as the Table's title.

The author must format (move the title at the other page).

 Answer: Fixed

Comment 3

Figure 7

Tiempo [s]

The author must check for spelling and typographical errors.

Answer:  Fixed

Comment 4

Line 826

The author must format the References according to journal's instructions (delete the gap - page 27) 

For all References (1 - 43)

1 Basu, P.;

The author must replace (add a .)

  1. Basu, P.;

Answer: All fixed. Also all reference are formatted and checked.
